# Multi-omics profiling of primary small cell carcinoma of the esophagus reveals RB1 disruption and additional molecular subtypes

Renda Li [1], Zhenlin Yang [1], Fei Shao[1,2], Hong Cheng[1], Yaru Wen[3], Sijin Sun[1], Wei Guo [1], Zitong Li[1], Fan Zhang[1], Liyan Xue[3], Nan Bi[4], Jie Wang[5], Yingli Sun[6], Yin Li[1], Fengwei Tan[1], Qi Xue[1,7], Shugeng Gao[1,7], Susheng Shi[3,8✉], Yibo Gao [1,7,8✉] & Jie He[1,7,8✉]

Primary small cell carcinoma of the esophagus (PSCCE) is a lethal neuroendocrine carcinoma. Previous studies proposed a genetic similarity between PSCCE and esophageal squamous cell carcinoma (ESCC) but provided little evidence for differences in clinical course and neuroendocrine differentiation. We perform whole-exome sequencing, RNA sequencing and immunohistochemistry profiling on 46 PSCCE cases. Integrated analyses enable the discovery of multiple mechanisms of *RB1* disruption in 98% (45/46) of cases. The transcriptomic landscape of PSCCE closely resembles small cell lung cancer (SCLC) but differs from ESCC or esophageal adenocarcinoma (EAC). Distinct gene expression patterns regulated by *ASCL1* and *NEUROD1* define two molecular subtypes, PSCCE-A and PSCCE-N, which are highly similar to SCLC subtypes. A T cell excluded phenotype is widely observed in PSCCE. In conclusion, PSCCE has genomic alterations, transcriptome features and molecular subtyping highly similar to SCLC but distinct from ESCC or EAC. These observations are relevant to oncogenesis mechanisms and therapeutic vulnerability.

[1] Department of Thoracic Surgery, National Cancer Center/National Clinical Research Center for Cancer/Cancer Hospital, Chinese Academy of Medical Sciences and Peking Union Medical College, Beijing, P. R. China. [2] Cancer Institute of the Affiliated Hospital of Qingdao University, Qingdao Cancer Institute, Qingdao, P. R. China. [3] Department of Pathology, National Cancer Center/National Clinical Research Center for Cancer/Cancer Hospital, Chinese Academy of Medical Sciences and Peking Union Medical College, Beijing, P. R. China. [4] Department of Radiation Oncology, National Cancer Center/National Clinical Research Center for Cancer/Cancer Hospital, Chinese Academy of Medical Sciences and Peking Union Medical College, Beijing, P. R. China. [5] Department of Medical Oncology, National Cancer Center/National Clinical Research Center for Cancer/Cancer Hospital, Chinese Academy of Medical Sciences and Peking Union Medical College, Beijing, P. R. China. [6] Central Laboratory, National Cancer Center/National Clinical Research Center for Cancer/Cancer Hospital & Shenzhen Hospital, Chinese Academy of Medical Sciences and Peking Union Medical College, Shenzhen, P. R. China. [7] State Key Laboratory of Molecular Oncology, National Cancer Center/National Clinical Research Center for Cancer/Cancer Hospital, Chinese Academy of Medical Sciences and Peking Union Medical College, Beijing, P. R. China. [8] These authors contributed equally: Yibo Gao, Susheng Shi, Jie He. ✉email: gaoyibo@cicams.ac.cn; shishusheng@sina.com; hejie@cicams.ac.cn

Primary small-cell carcinoma of the esophagus (PSCCE) is one of the deadliest neuroendocrine malignancies, featuring rapid progression, high metastasis propensity, and dismal prognosis. The 5-year survival rates of limited-stage patients were ~10%[1–3] and were practically zero for extensive-stage patients[4,5]. The prevalence of PSCCE has been increasing in the recent decade[5], estimated to account for 0.5–2.8% of all esophageal malignancies[6–8]. However, no consensus on standardized treatment for PSCCE has been reached at present, and all current strategies showed limited capacity to improve prognosis.

Diagnosis of PSCCE requires appropriate morphology and positive immunohistochemistry (IHC) staining of neuroendocrine markers including CD56, Synaptophysin, and Chromagranin A[9]. PSCCE has long been observed to have different morphology[10], disease course[2,3,8] and IHC staining[9] from esophageal squamous cell carcinoma (ESCC) or esophageal adenocarcinoma (EAC), two predominant histologies of esophageal cancer. Indeed, most management strategies for PSCCE were extrapolated from those designed for small cell lung cancer (SCLC) based on their similar histology.

To date, there was no comprehensive multi-omics study of PSCCE. One previous whole-exome sequencing (WES) study of 55 PSCCE revealed disruption of putative tumor-suppressors including TP53 (80%), RB1 (27%), and NOTCH1 (24%), claiming that PSCCE genetically resembled ESCC[11]. However, the reported mutation rate of RB1 was significantly lower than that reported in SCLC (93%)[12]. Knowledge of transcriptomic features, molecular subtyping, and therapeutic targets is still lacking.

Integrated genomics and transcriptomics studies of SCLC converged on four distinct molecular subtypes each with unique susceptibilities to different targeted therapies[13–15], which was informative to future researches and clinical trials. These studies of SCLC inspired several studies on promising therapeutic targets, including the inhibition of Aurora kinase[14,16] and targeting metabolism addition of specific SCLC subtype[17].

To provide a more comprehensive understanding of the biology of PSCCE, we perform genomic, transcriptomic, and immune profiling of 46 PSCCE screened from 7539 consecutive esophageal cancer patients treated in a single institute. By integrating multi-omics studies, we discover an array of molecular features distinguishing PSCCE from ESCC or EAC, several of which may translate into therapeutic targets. We identify the nearly universal disruption of RB1 by multiple mechanisms, confirming the fundamental role for RB1 in PSCCE. We identify two molecular subtypes of PSCCE, each with unique genomic and transcriptomic features. The tumor microenvironment (TME) of PSCCE is featured by insufficient T-cell infiltration, which might be induced by some intrinsic expression programs. These findings deepen our understanding of PSCCE and may translate into clinical impact.

## Results

**Patients and study overview.** We retrospectively screened 7539 consecutive esophageal cancer patients treated in the Department of Thoracic Surgery from 2006 to 2017 and identified 65 PSCCE cases (0.9%). After intensive quality evaluation, 46 cases with complete medical records and sufficient specimens for sequencing were included in our study (Supplementary Data 1).

We performed WES on 46 pairs of tumor and matched normal sample, including 13 pairs of fresh-frozen (FF) samples and 33 pairs of formalin-fixed, paraffin-embedded (FFPE) samples. The average coverages for FF samples were 333× and 172× for tumors and normal samples, respectively. The average coverages for FFPE samples were 314× and 166× for tumors and normal samples, respectively. Sequencing coverages were comparable

between FF samples and FFPE samples (tumors, $P = 0.647$; normal samples, $P = 0.454$, all by the Wilcoxon rank-sum test).

We performed RNA Exome Access sequencing (RNA-seq) on FFPE samples of 38 tumors and 23 matched normal esophageal samples, achieving an average read depth of 86.7 million reads per sample.

**Mutational burden and signatures.** We identified 4918 somatic mutations by WES, including 1200 synonymous single-nucleotide variants (SNVs), 3511 nonsynonymous SNVs and 207 indels (Supplementary Data 2), with an average of 81 nonsynonymous mutations per tumor (range: 16–190), corresponding to an average nonsynonymous mutation rate of 2.31 mutations per megabase (Mb).

We observed that FFPE samples demonstrated significantly higher mutational burden than FF samples (overall mutation rate: 3.36/Mb and 2.28/Mb for FFPE and FF samples, respectively, $P = 0.00628$, Wilcoxon rank-sum test; nonsynonymous mutation rate: 2.54/Mb and 1.72/Mb for FFPE and FF samples, respectively, $P = 0.0044$, Wilcoxon rank-sum test). The proportion of low allele frequency (AF) mutations (AF < 0.05) in FFPE samples (887/ 3881, 23%) was significantly higher than that in FF samples (164/ 1037, 16%, $P = 1.14 \times 10^{-6}$, Chi-square test), consistent with previous reports that FFPE samples were enriched for low-AF nucleotide transitions induced by fixation[18]. The proportions of nonsynonymous mutations in all mutation were comparable between FFPE (75.6%, 2933/3881) and FF samples (75.7%, 785/ 1037, $P = 0.9656$, Chi-square test).

The nonsynonymous mutation rate in this study was significantly higher than that reported by Wang et al.[11] (2.12/ Mb, $P = 0.0225$, Wilcoxon rank-sum test), owing to the higher sequencing depth. The nonsynonymous mutation rate of PSCCE was significantly lower than those of ESCC[19–21] (3.15/Mb, $P = 0.00021$), EAC[21] (5.16/Mb, $P = 1.11 \times 10^{-7}$) and SCLC[12] (8.62/ Mb, $P < 2.2 \times 10^{-16}$, all by the Wilcoxon rank-sum test), ranking medially among cancers sequenced by The Cancer Genome Atlas (TCGA, Supplementary Fig. 1a) project.

We observed a high frequency of C>T substitutions, comprising 51.1% (2407/4711) of all SNVs (including 1200 synonymous SNVs and 3511 nonsynonymous SNVs), especially in the NpCpG trinucleotide context (Fig. 1a). C>T substitution was the most common SNVs in both FF and FFPE samples; however, the proportion of C>T substitutions was higher in FFPE samples (53.1%) than in FF samples (43.1%), consistent with previous reports of C>T substitution enrichment in FFPE samples resulted from deamination during fixation[22]. The overall mutational spectrum was highly similar to that reported by Wang et al.[11] (cosine similarity: 0.97, Supplementary Fig. 1b), while similar to EAC[21] (0.89) and ESCC[21] (0.85) to a lesser extent and quite different from SCLC[12] (0.62).

In the integrated analysis of mutational signatures of PSCCE ($n = 101$, combined with 55 cases reported by Wang et al.[11], hereinafter referred to as the "combined cohort"), SCLC ($n = 110$, ref. [12]), EAC ($n = 88$, TCGA), and ESCC ($n = 96$, TCGA), we identified five principal mutational signatures, E1–E5 (Fig. 1b), which were highly similar to Catalogue Of Somatic Mutations In Cancer (COSMIC) signatures 1, 13, 4, 16, and 17, respectively. We did not discover any de novo signature unique to small-cell carcinomas. The proportions of signatures varied among cancer types (Fig. 1c), reflecting the effects of both exposure and intrinsic tumorigenesis mechanism. Endogenous 5-methylcytosine deamination-associated Signature E1 was the predominant mutational signature in PSCCE. Signature E4 was recently reported to be associated with alcohol drinking[23]. Consistent with this observation, Signature E4 contribution was significantly higher in regular

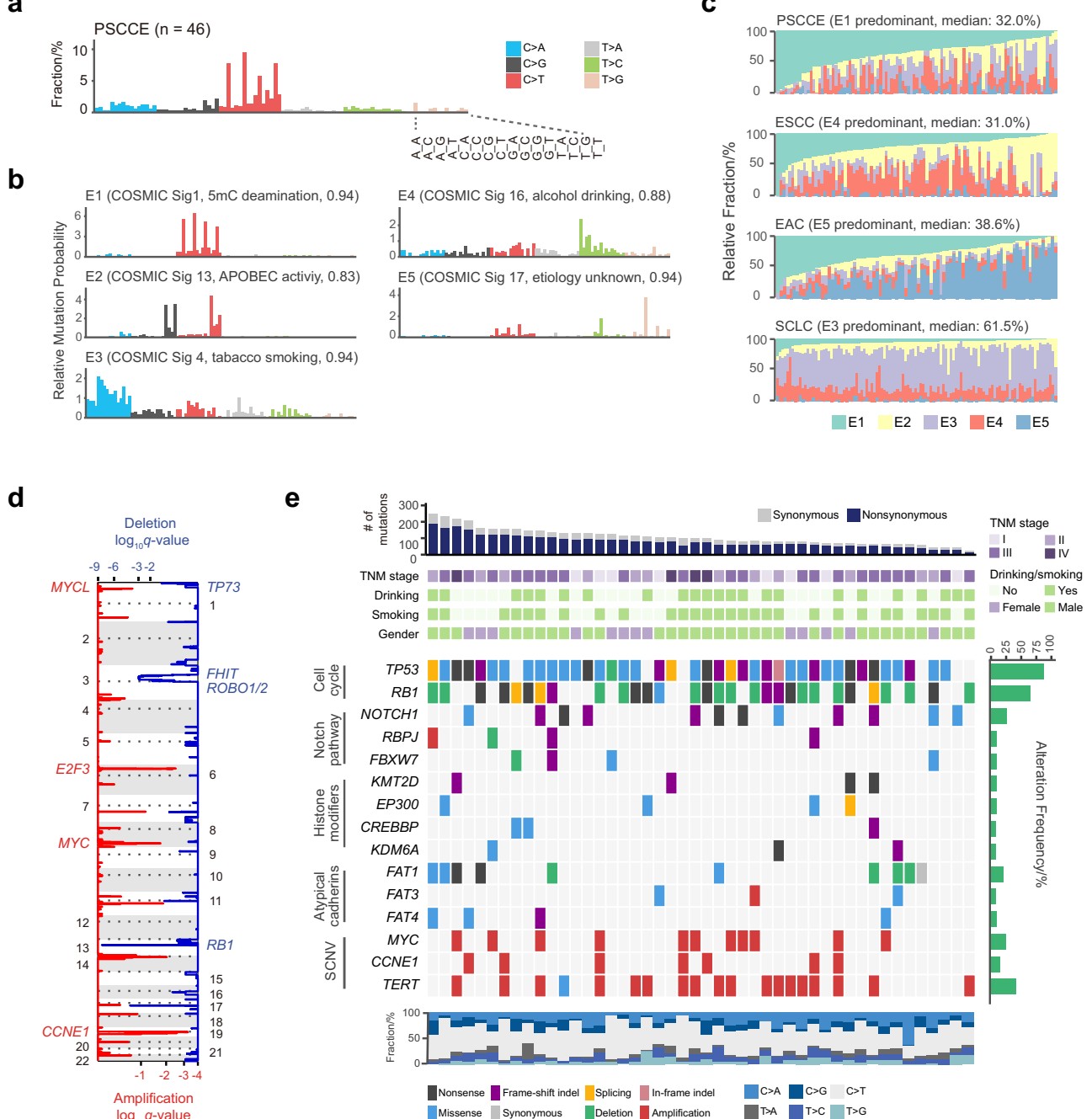

**Fig. 1 Genomic alterations detected by WES in PSCCE. a** Mutational spectrum of PSCCE. Substitutions are plotted in different colors with their context arranged in the denoted order. **b** Five mutational signatures, denoted as E1–E5, were identified in the integrated analysis of PSCCE, ESCC, EAC, and SCLC. COSMIC signatures showing high similarity, its etiology and cosine similarity score are shown. **c** The proportions of signature E1–E5 in each sample of PSCCE, ESCC, EAC, and SCLC tumors. The predominant signature in each cancer is shown. **d** Recurrent somatic copy-number variations in PSCCE. Amplifications and deletions are plotted in red and blue, respectively. **e** Landscape of somatic alterations in PSCCE. Somatic alterations of each gene (row) in each tumor (column) are plotted as a heatmap according to the color legend below. Samples are arranged by the number of mutations (top panel). Clinical parameters of each patient are shown below. Alteration frequencies are shown in the right panel. The proportion of single-nucleotide substitutions in each sample is shown in the bottom panel.

drinkers of PSCCE patients (median 25.6%) compared to never or occasional drinkers (median 13.0%, $P = 0.039$, Wilcoxon rank-sum test).

**Somatic copy-number variations (SCNVs).** We performed data cleaning and applied stringent thresholds to identify solid somatic copy-number variations (SCNVs) from the noisy background of FFPE WES data (Source Data 1).

GISTIC2.0 algorithm[24] identified significant deletions (Fig. 1d and Supplementary Data 3) of 13q14 (harboring *RB1*) and 3p12-14 (harboring *FHIT* and *ROBO1*), both of which were also recurrently deleted in SCLC[12]. We also observed amplifications of canonical cancer genes, including *CCNE1* on 19q12 and *MYC* on 8q24.

We validated the copy numbers of 102 loci in 34 samples (including 12 FF and 22 FFPE; *MYC*, *TERT*, and *SOX4* loci for each sample, Supplementary Data 4) by quantitative polymerase chain reaction (PCR). Thirty five of 36 (97%) loci in FF samples and 61 of 66 (92%) loci in FFPE samples showed consistent results between WES and quantitative PCR.

Several genes in cell-cycle pathway and receptor tyrosine kinase pathway showed remarkably different frequency of SCNV between PSCCE and esophageal cancers. *CCND1* amplification was observed in 33–54% of ESCCs[19,21] and in only 4% of PSCCEs ($P = 1.06 \times 10^{-7}$, Fisher's exact test). *CDKN2A* deletion was reported in up to 66% of ESCCs[19,21] but was only observed in 4% of PSCCEs ($P = 2.46 \times 10^{-6}$, Fisher's exact test). *ERBB2* was amplified in 17–27% of EACs[21,25] but in none of PSCCEs ($P = 7.36 \times 10^{-5}$, Fisher's exact test).

**Recurrently mutated genes and pathways.** The MutsigCV algorithm[26] identified three significantly mutated genes (SMGs, $P < 0.05$ and $q < 0.1$, Fig. 1e, Supplementary Data 5): *TP53*, *RB1*, and *NOTCH1*. Two genes were identified with significant mutation clusters ($P < 0.05$ and $q < 0.1$, Supplementary Data 5): *TP53* and *NOTCH1*. In the combined cohort of 101 PSCCE patients, no more SMGs were identified, while two more genes were identified with significant mutation clusters: *EP300* and *FBXW7* (Supplementary Fig. 1c).

Sequencing of FFPE samples was reported to have comparable performance in detecting driver mutations and actionable events to sequencing of FF samples[18]. We performed Sanger sequencing of somatic mutations in *TP53* ($n = 54$), *RB1* ($n = 19$), and *NOTCH1* ($n = 15$, Supplementary Data 6). Twenty-four of 24 loci (100%) in FF samples and 55 of 64 loci (86%) in FFPE samples were successfully amplified by PCR. All loci that were amplified were validated by Sanger sequencing. We further validated seven of nine loci that suffered PCR amplification failure by confirming expression of somatic mutations in RNA-seq reads (Supplementary Fig. 2).

In total, 54 somatic mutations (43 nonsynonymous SNVs, 2 synonymous SNVs and 9 indels) of *TP53* were observed, affecting 85% (39/46) of all tumors, similar to the previous report by Wang et al.[11] (44/55, $P = 0.608$, Fisher's exact test). Thirty-four percent (18/52) of the nonsynonymous mutations of *TP53* were nonsense, frame-shifting indels or splice site mutations truncating the protein.

We identified 22 genes with established roles in cancers (the Cancer Gene Census, ref. [27]) among 97 genes that were nonsynonymously mutated in at least three tumors. We observed frequent mutations in histone modifier genes. Histone acetyl-transferase *EP300* and *CREBBP* were mutated in 9% (4/46) and 7% (3/46) of cases, respectively. Two out of four *EP300* mutations and all four *CREBBP* mutations affected the histone acetyltransferase domain. COMPASS-like complex components *KMT2D* and *KDM6A*, which play important roles in modifying histone methylation, were mutated in 4 and 3 tumors, respectively. Mutations of *EP300*, *CREBBP*, *KMT2D*, and *KDM6A* were largely mutually exclusive, affecting 26% (12/46) of all PSCCEs (Fig. 1e). *FAT* atypical cadherin family, including *FAT1*, *FAT3*, and *FAT4*, which are considered tumor-suppressive in numerous cancers[28], were mutated in a total of ten cases.

**Nearly universal disruption of *RB1* by multiple mechanisms.** We identified 19 somatic mutations of *RB1* affecting 16 (34.8%) cases (Fig. 2a). The mutation frequency of *RB1* was comparable to that reported by Wang et al.[11] (15/55, 27.3%, $P = 0.5166$, Fisher's exact test). Seventy-nine percent (15/19) of *RB1* mutations were

truncating, including 9 nonsense, 3 frame-shifting indels, and 3 splice site mutations.

SCNV analysis discovered that 14 (30.4%) tumors harbored homozygous deletions affecting *RB1*, including 13 homozygous deletions affecting only part of but not the whole *RB1* locus (Supplementary Data 7, hereinafter referred to as "*RB1* exon deletions"), and one homozygous deletion of whole *RB1* locus in PSCCE_79T. *RB1* exon deletions affected a median of 11 exons (range: 1–25) and had a median length of 52.5 kilobase (kb, range: 0.2–2828 kb). *RB1* exon deletions left the unaffected exons intact and were mutually exclusive to somatic mutations ($P = 0.00117$, hypergeometric test). According to the SCNV results, we designed series of PCR primers to locate breakpoints of *RB1* exon deletions, consolidating the bioinformatics SCNV findings (Fig. 2b and Supplementary Data 7). *RB1* exon deletions occurred at a distinguishably high frequency (28.3%) in PSCCE, remarkably higher than that of 110 whole-genome sequenced SCLC[12] (8.2%, $P = 0.002$, Fisher's exact test) and 508 whole-genome sequenced ESCCs[29] (0.79%, $P = 2.43 \times 10^{-13}$, Fisher's exact test). *RB1* exon deletions were not thoroughly described in the previous WES profiling of PSCCE[11].

We further integrated RNA-seq and IHC profiling to discover additional *RB1*-disrupting events. We found that 28 of 38 RNA-sequenced tumors harbored splicing abnormalities of *RB1* mRNA, including exon skipping ($n = 21$), gene fusion ($n = 2$), formation of new splice site ($n = 1$), and disrupted expression of 3′-terminal exons ($n = 4$, Supplementary Data 8). RNA-seq recapitulated splicing abnormalities of *RB1* mRNA that are the deduced consequences of exon deletions and splicing site mutations (Fig. 2b, c and Supplementary Fig. 3a, b). Eight tumors with splicing abnormality in RNA-seq were otherwise *RB1* "wild type" in WES (Supplementary Fig. 3c), suggesting alterations undetectable by WES.

Five tumors (11%) stained positive for *RB1* protein (Rb) by IHC (Supplementary Data 8). However, deleterious abnormalities were observed in 4 of them. In PSCCE_10T and PSCCE_13T, in-frame exon deletions excluded exons encoding >400 amino acids (Fig. 2c) but retained C-terminal epitope for IHC antibody recognition. In PSCCE_10T, two alleles of *RB1* suffered different deletions: one allele suffered exons 1–17 deletion (encoding 499 amino acids) and formed in-frame fusion with the upstream *ITM2B*; the other allele suffered frame-shifting deletion of exons 3–18 (Fig. 2c). Strong Rb staining was observed in cytoplasm but not nucleus of PSCCE_10T. Two tumors, namely PSCCE_32T and PSCCE_56T, were wild type by WES and stained positive for Rb. However, RNA-seq revealed in-frame exon skipping of *RB1* mRNA in PSCCE_32T (exons 3–17 skipping, encoding 477 amino acids) and PSCCE_56T (exons 14–17 skipping, encoding 121 amino acids, Supplementary Fig. 3d), which severely disrupted Rb function. Notably, Rb was completely lost in four cases with no abnormalities observed by WES or RNA-seq (Fig. 2a and Supplementary Fig. 3e), indicating mechanisms disrupting *RB1* mRNA translation or protein stability.

By integrating WES, RNA-seq, and IHC results, we discovered disruption of *RB1* in 45 (98%) cases. Only one tumor, PSCCE_33T, stained positive for Rb and showed no abnormality in WES; we had no RNA to test in-frame exon skipping in PSCCE_33T. Integrative analysis identified substantially more *RB1* disruption events than did the single WES performed by us or by Wang et al.[11]. *RB1* disruption frequency was remarkably higher than ESCC[21] (10%, 9.8-fold, $P < 2.2 \times 10^{-16}$, Fisher's exact test), EAC[21] (3.4%, 28.8-fold, $P < 2.2 \times 10^{-16}$, Fisher's exact test) and was comparable to SCLC[12] (93%).

**PSCCE has a transcriptome highly similar to SCLC.** We next turned to the transcriptomic landscape of PSCCE (Source Data 2).

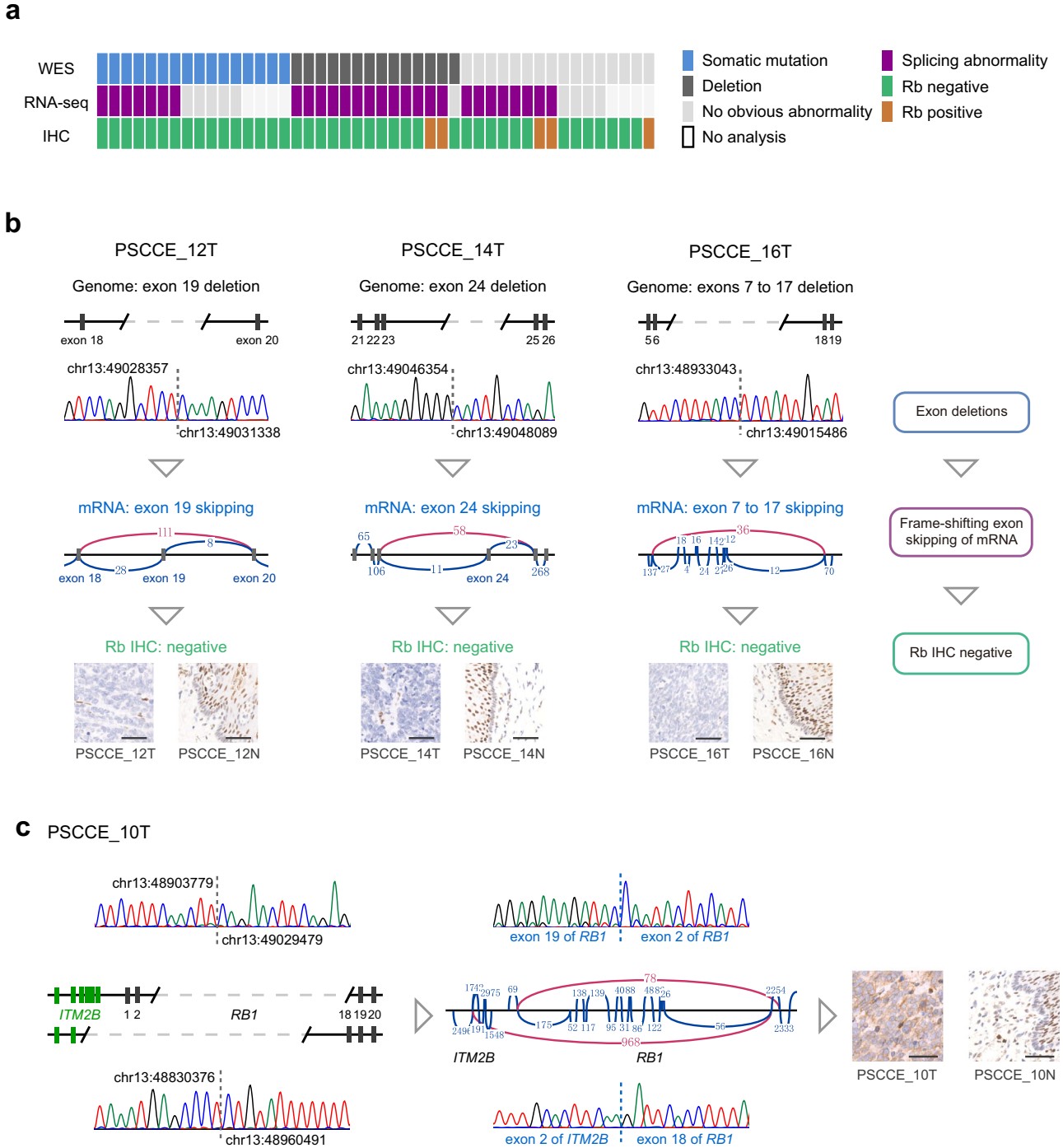

**Fig. 2 *RB1* disruption by multiple mechanisms. a** Schematic summary of *RB1* disruption events observed in each sample (column) by different methods (row). The rightmost sample is PSCCE_33T, the only tumor positive for Rb, yet not evaluated by RNA-seq. **b** (From top to bottom) Schematic plot of exon deletions, Sanger sequencing validation of exon deletion breakpoints, sashimi plot of subsequent splicing abnormalities in *RB1* mRNA and Rb IHC of three representative tumors. Schematic summary of disrupting mechanism is shown on the right. Black rectangles represent *RB1* exons and gray dashed line represent deleted genomic regions. Green, red, blue, and black peaks in Sanger sequencing chromatograms represent bases A, T, C, and G, respectively. Genomic coordinates are in hg19 assembly. Curves in sashimi plot represent reads spanning exon junction with numbers of reads denoted. Abnormal exon junctions are plotted in red. Rb IHC of matched normal sample is provided as control. Scale bar: 50 μm. **c** Schematic plot and Sanger sequencing validation of exon deletions (left), sashimi plot and validation of abnormal mRNA exon junctions (middle) and Rb IHC staining (right) of PSCCE_10T. Two alleles of *RB1* are plotted separately to show different ranges of exon deletions. Exons of *ITM2B* are plotted in green. Sanger sequencing chromatograms and sashimi plots are plotted in the same way described above. Scale bar: 50 μm.

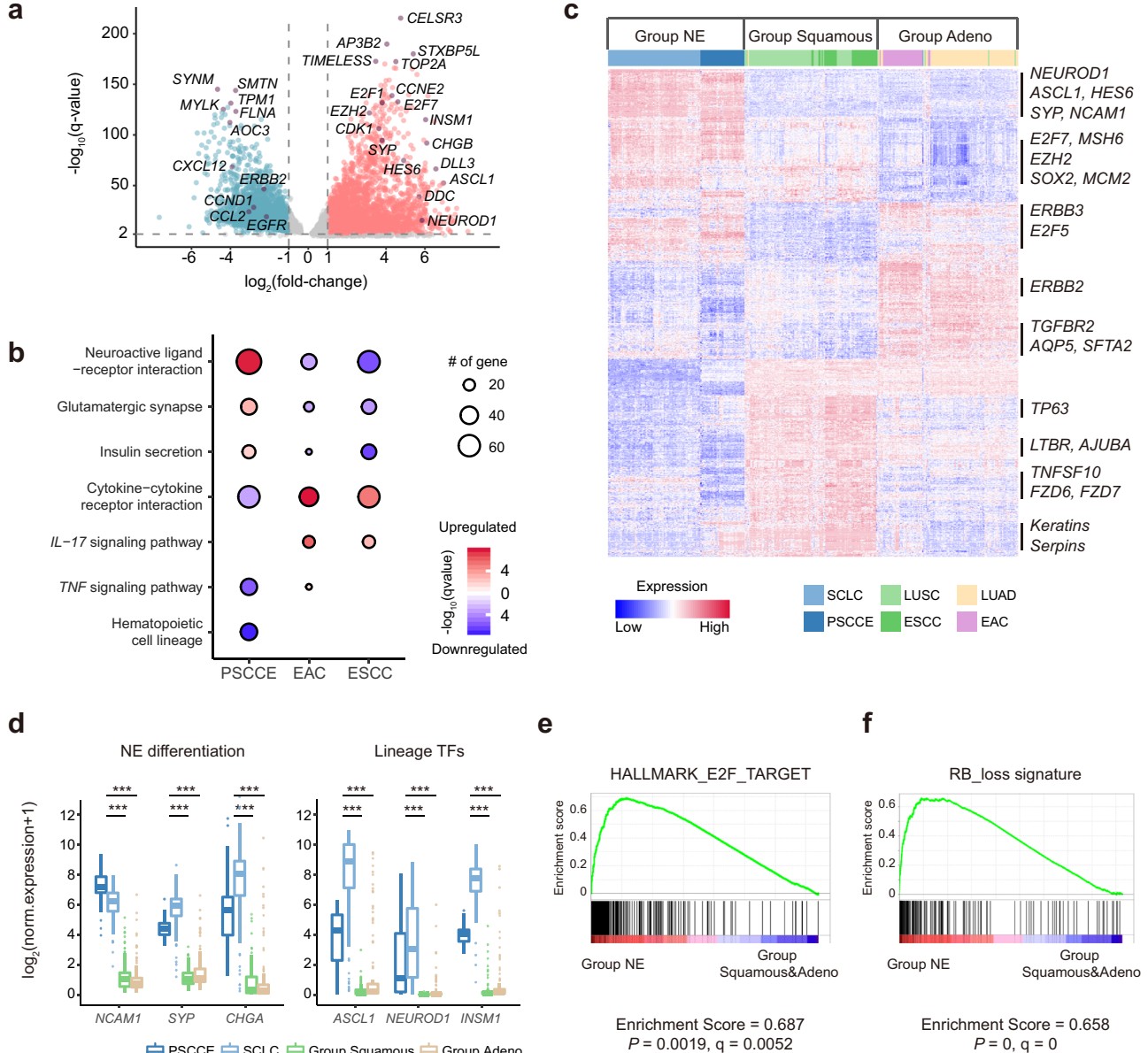

**Fig. 3 The transcriptome landscape of PSCCE. a** Volcano plot of differentially regulated genes (DEGs). Up- and downregulated DEGs are plotted in red and cyan, respectively. Key genes are plotted in purple with symbols annotated. Source data are provided as a Source Data file. **b** Pathways that were largely discordant between PSCCE, EAC, and ESCC. **c** Heatmap of gene expressions of three principle clustering groups. Key signature genes associated with clustering are marked on the right side. **d** Expression of neuroendocrine differentiation marker genes and lineage transcription factors across three groups. PSCCE and SCLC samples are plotted separately to show that there was no hijacking of either cancer by the other. The upper bound, centerline, and lower bound of boxplot represent the 75 percentile, the median and the 25 percentile of data; the upper and lower whiskers extend to the largest and smallest value within 1.5 times of interquartile range (IQR) from corresponding bound. Data beyond the whiskers are plotted as outlier dots. ***$P < 0.001$, by the Wilcoxon rank-sum test. **e** GSEA revealed that $E2F$ target genes were specifically expressed in Group NE. **f** GSEA revealed that $RB1$-loss associated genes were specifically expressed in Group NE.

We compared the gene expressions of 38 tumors against 23 matched normal esophageal samples to identify differentially expressed genes (DEGs, Source Data 3). DEG analysis revealed significant upregulation of genes involved in DNA replication, cell cycle, and neuroendocrine differentiation (Fig. 3a and Supplementary Fig. 4a) and downregulation of genes related to cell adhesion in PSCCE. By enrichment analysis, we discovered several pathways that were largely distinguishable between PSCCE and esophageal cancers (Fig. 3b). Neuroactive ligand-receptor interaction pathways were specifically upregulated in PSCCE, which was consistent with its neuroendocrine differentiation. Upregulated DEGs of EAC and ESCC were significantly enriched for

immune response pathways. In contrast, these pathways were attenuated in PSCCE.

Given the prominent neuroendocrine differentiation of PSCCE and its divergence from EAC and ESCC, we sought to describe relationship with its counterpart in the lung, SCLC. We collected transcriptomic profiles of SCLC[12], ESCC[21], EAC[21], lung squamous cell carcinoma (LUSC)[30], and LUAD[31] (Supplementary Fig. 4b and Supplementary Data 9). Iterations of unsupervised clustering yielded three principal groups that showed distinct gene expression patterns ("Group NE," "Group Squamous," and "Group Adeno," Fig. 3c). Consensus clustering demonstrated that the transcriptome landscape of PSCCE was

highly similar to SCLC: all PSCCEs and all but one SCLCs clustered together into Group NE.

We identified signature genes highly and specifically expressed in each group against other groups (Supplementary Data 10). Group NE signature genes included several neuroendocrine markers including *NCAM1*, *SYP*, and *CHGA* (Fig. 3d). These genes had vanishingly low expressions in other groups. Genes associated with squamous cell and gland differentiation were among signature genes specifically expressed in Group Squamous and Group Adeno, respectively (Supplementary Fig. 4c).

We next turned to transcription regulation networks of signature genes. Several neuroendocrine lineage transcription factors including *ASCL1*, *NEUROD1*, and *INSM1* were among Group NE signature genes (Fig. 3d). Group NE signature genes were also significantly enriched for targets of the *E2F* family (Supplementary Fig, 4d). Gene set enrichment analysis (GSEA) revealed significant enrichment of *E2F* targets in Group NE (Fig. 3e), consistent with relief of *E2F* suppression by *RB1* loss observed in both PSCCE and SCLC. Group Squamous signature genes were regulated by *TP63*, consistent with its squamous cell differentiation.

The three groups also showed differential activation of oncogenic signaling pathways. The *RB1* depletion-associated signature[32] ("RB-loss signature") was specifically enriched in Group NE tumors, consistent with universal *RB1* disruption in both PSCCE and SCLC (Fig. 3f). *EGFR* signaling pathway was preferentially activated in Group Squamous and Group Adeno (Supplementary Fig. 4e), consistent with observations of *EGFR* signaling activation by activating mutations or overexpression of *EGFR* in lung and esophageal cancers[31,33].

Taken together, the distinguishable expression pattern, regulation network, and oncogenic pathway activation encapsulated that PSCCE was different from EAC and ESCC but highly similar to SCLC. PSCCE should not be viewed as a neuroendocrine variant of ESCC but as a distinct entity.

**Gene expression pattern reveals two subtypes of PSCCE**. We further looked into the transcriptome of PSCCE alone to decipher the heterogeneity within PSCCE, which would be masked when compared with strongly divergent cancers.

Unsupervised consensus clustering of 38 PSCCE tumors yielded two molecular subtypes with distinct gene expression patterns (Fig. 4a and Supplementary Data 11). The first subtype comprised nine tumors stably clustered together. The other 29 tumors showed unstable clustering but seldom clustered with the first subtype and were hence considered collectively. We found lineage transcription factors *NEUROD1* and *ASCL1* within signature genes (Supplementary Data 12) of each group; they were also among the most differentially expressed genes across the two subtypes (Fig. 4b). According to these observations, we named the 9 *NEUROD1*^high tumors as subtype PSCCE-N, and the other 29 *ASCL1*^high tumors as subtype PSCCE-A. GSEA revealed that target genes of *NEUROD1* and *ASCL1*[13] were preferentially expressed in PSCCE-N and PSCCE-A tumors, respectively (Fig. 4c), confirming the regulatory roles of *ASCL1* and *NEUROD1*. The inverse expression pattern of *ASCL1* and *NEUROD1* in two subtypes was validated by IHC (Fig. 4d). Several neuroendocrine marker genes including *GRP*, *DDC*, and *SSTR2* also showed remarkable differences between two subtypes (Fig. 4b).

We screened for genomic alterations associated with this subtyping and found that PSCCE-N subtype was associated with amplification of multiple segments on chromosome 8q, including region harboring cancer gene *MYC* (Supplementary Fig. 5a). The fraction of patients with *MYC* amplification was significantly

higher in PSCCE-N (56%, 5/9) than in PSCCE-A (17%, 5/29, $P = 0.036$, Fisher's exact test). However, the expression levels of *MYC* and *MYCL* mRNA showed no significant difference between two subtypes (Supplementary Fig. 5b). The PSCCE-N subtype had a significantly higher *MYCN* level than PSCCE-A ($P = 0.0012$, Wilcoxon rank-sum test, Supplementary Fig. 5b).

We also observed that PSCCE-N patients had significantly worse prognoses when compared with PSCCE-A patients (hazard ratio (HR): 2.44, 95% confident interval (CI): 1.04–5.69, $P = 0.0394$, log-rank test, Fig. 4e). Other clinical features, including gender, age at diagnosis, T stage, N stage, and TNM stage presented no significant difference between two subtypes (Supplementary Data 13).

A synthesis of recent studies of SCLC converged on four molecular subtypes—SCLC-A, SCLC-N, SCLC-P, and SCLC-Y, as featured by the differential expression of lineage transcription factors *ASCL1*, *NEUROD1*, *POU2F3*, and *YAP1*, respectively[15]. We found that PSCCE subtypes highly resembled the SCLC-A and SCLC-N subtypes. Gene sets associated with SCLC-A and SCLC-N tumors[13] were specifically expressed in PSCCE-A and PSCCE-N tumors, respectively (Supplementary Fig. 5c). The SCLC-N subtype was reported to associate with *MYC* amplification[14,34] and to rapidly metastasize and relapse in human and murine model[14,34], consistent with our findings that PSCCE-N subtype was associated with MYC amplification and poorer prognoses.

*POU2F3* and *YAP1* levels in the 38 RNA-sequenced PSCCEs were relatively low compared to *ASCL1* and *NEUROD1* and did not appear to be a selective master regulator (Supplementary Fig. 6). Given the prominent *ASCL1*- or *NEUROD1*-associated expression patterns observed, the evidence obtained in the present study was insufficient to confirm a *POU2F3*- or *YAP1*-dominated subtype of PSCCE.

**Notch pathway inactivation in PSCCE**. Notch signaling was considered tumor-suppressive in SCLC[12]. In PSCCE, we also observed low activity of Notch signaling, as characterized by significant downregulation of Notch receptors and effector, and upregulation of Notch antagonists (Fig. 5a).

The inhibition of Notch signaling was observed on multiple levels. (a) *NOTCH1* was mutated in 26% (12/46) of cases. Sixty-four percent (9/14) of *NOTCH1* nonsynonymous mutations were frame-shifting deletions and nonsense mutations truncating the protein far N-terminal to its transactivation domain, and were thus considered inactivation. Missense mutations of *NOTCH1* showed significant local clustering in extracellular epidermal growth factor-like repeats important for ligand binding (Fig. 5b). Consistent with its tumor-suppressive role, mutations in *NOTCH1-4* were significantly associated with a poorer prognosis in the combined cohort (HR 1.67, 95% CI 1.01–2.69, $P = 0.046$, Fig. 5c). (b) The relatively mutation-sparse *NOTCH2* and *NOTCH3* genes were identified as downregulated DEGs when compared to matched normal samples (Fig. 5d, detailed *P*, *q*, and log2FC value in Source Data 3). (c) Transcriptomic profiling also revealed inhibitory ligands of Notch pathway *DLK1* and *DLL3* as upregulated DEGs (Fig. 5d and Source Data 3). (d) *HES6*, a member of basic helix-loop-helix transcription factor, which binds and inhibits the major Notch pathway effector *HES1*[35], was overexpressed in PSCCE (Source Data 3).

The downregulation of Notch receptors and overexpression of Notch antagonists were also observed in tumors with wild-type Notch receptors (Fig. 5d), indicating a constitutive suppression of Notch signaling. Consequently, pro-neural TFs *ASCL1* and *NEUROD1*, whose expression was inhibited by Notch signaling pathway[36], were significantly upregulated.

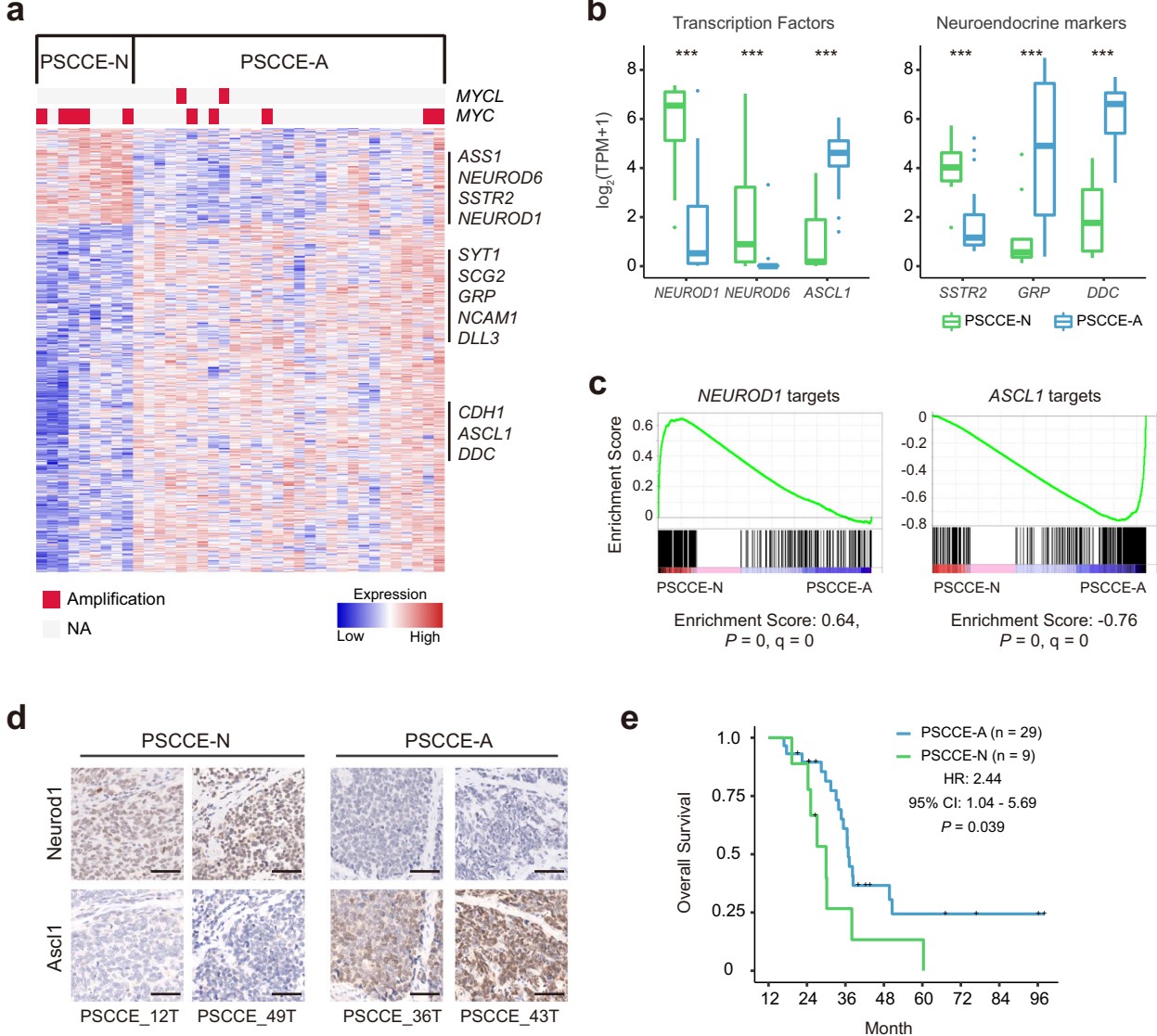

**Fig. 4 PSCCE had two molecular subtypes. a** Gene expression heatmap of PSCCE-A and PSCCE-N subtypes identified by unsupervised consensus clustering. Amplifications of MYC family are shown on the top of the heatmap. Selected key genes related to subtyping are shown on the right side. NA: no significant SCNV observed. Source data are provided as a Source Data file. **b** Expressions of lineage transcription factor and neuroendocrine marker genes that were significantly differentially expressed across two subtypes. The upper bound, centerline, and lower bound of boxplot represent the 75 percentile, the median and the 25 percentile of data; the upper and lower whiskers extend to the largest and smallest value within 1.5 times of IQR from corresponding bound. Data beyond the whiskers are plotted as outlier dots. ***$P < 0.001$, by the Wilcoxon rank-sum test. **c** GSEA revealed that target genes of *NEUROD1* and *ASCL1* were preferentially expressed in PSCCE-N and PSCCE-A subtype. **d** IHC of Neurod1 and Ascl1 showed inverse expression pattern in PSCCE-N and PSCCE-A subtypes. Scale bar: 50 μm. **e** Kaplan–Meier plot of the overall survival of patients from two subtypes.

**PSCCE features a T-cell excluded tumor microenvironment.** In DEG pathway enrichment analysis, we observed downregulation of immune response pathways in PSCCE. To further describe TME in PSCCE, we used computational methods to evaluate abundance of infiltrating immune cells.

We first applied single-sample GSEA (ssGSEA) projection of immune cell signatures collected from the literatures ($n = 19$, Supplementary Data 14), including signatures of T cells, B cells, macrophages and granulocytes, on gene expression profiles of PSCCE, SCLC, LUAD, LUSC, EAC, and ESCC (see "Method" and Source Data 4). We found that PSCCE had significantly lower T cells and CD8 T cells scores than EAC and ESCC (Fig. 6a). Both organ-of-origin and histology had significant impacts on the immune milieu: esophageal cancers generally had significantly fewer infiltrating T cells than lung cancers, consistent with

previous reports[37]; and small-cell carcinomas had significantly lower T cell and CD8 T-cell abundance compared to tumors with other histologies arising from same organ (Fig. 6a and Supplementary Fig. 7a, two-way ANOVA test). Enumeration of immune infiltrates using a deconvolution method (CIBERSORT, ref. [38]) showed similar trend: EAC and ESCC had lower abundance of infiltrating T cells and CD8 T cells than LUAD and LUSC, while PSCCE had significantly even lower abundances than ESCC and EAC (both $P < 0.001$, Wilcoxon rank-sum test, Fig. 6b). We also quantified T-cell abundance using recent single-cell RNA-seq (scRNA-seq) derived cell type signature[39]. The Overall Expression (OE) scores for both T cells and CD8 cytotoxic T cells were significantly lower in PSCCE and SCLC than in corresponding non-neuroendocrine malignancies (Fig. 6c).

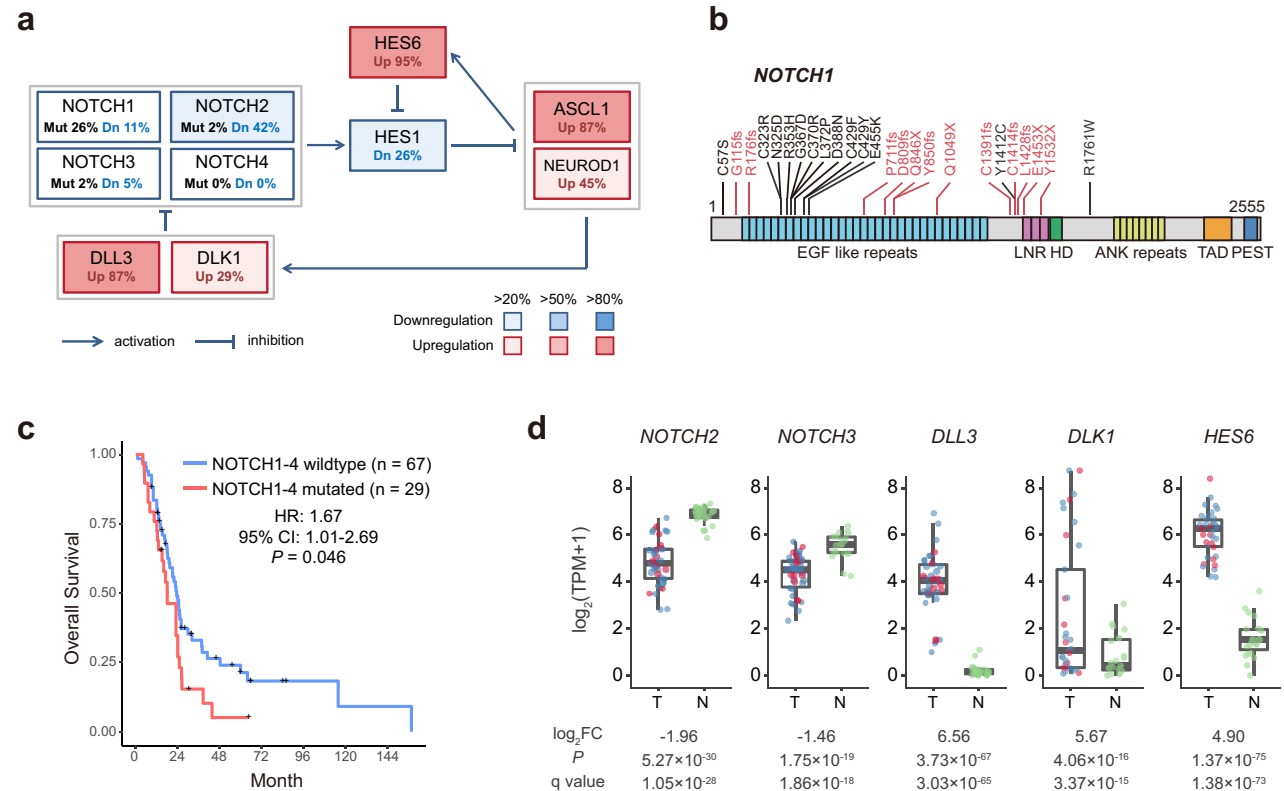

**Fig. 5 Notch pathway was inactivated in PSCCE. a** A schematic plot of Notch pathway dysregulation in PSCCEs. Up: upregulated, defined by having expression higher than three times of 95% quantile of 23 normal esophageal samples; Dn: downregulated, defined by having expression lower than one-third of 5% quantile of 23 normal esophageal samples; Mut: mutated. The proportions of tumors showing dysregulation are denoted as percentages, which are also indicated by color intensities according to the color legend. **b** Mutations of *NOTCH1* were enriched in extracellular EGF-like repeats. Truncating mutations are plotted in red. **c** Kaplan–Meier plots of overall survival of PSCCE patients. Patients with mutated *NOTCH1-4* had significantly poorer prognoses. **d** The dysregulation of Notch pathway components was observed regardless of Notch receptor status. The red dots represent tumors harboring mutations in Notch receptors (*NOTCH1-4*) and blue dots represent tumors with wild-type Notch receptors. Log$_2$(foldchange), *P* and *q* values determined by DESeq2 are shown below. T tumors. N normal samples. The upper bound, centerline, and lower bound of boxplot represent the 75 percentile, the median, and the 25 percentile of data; the upper and lower whiskers extend to the largest and smallest value within 1.5 times of IQR from corresponding bound. Data beyond the whiskers are plotted as outlier dots. Source data are provided as a Source Data file.

Most solid tumors could be categorized into three immune phenotypes: inflamed, excluded, and desert[40,41]. Consistent with the computational analysis, IHC staining of CD8A revealed that 85% (23/27, Supplementary Data 15) of PSCCEs were CD8 T-cell "excluded", in which CD8 T cells failed to infiltrate into tumor parenchyma and aggregated in the surrounding stroma instead (Fig. 6d and Supplementary Fig. 7b). A recent study showed that the exclusion of T cell was induced by certain programs expressed by malignant cells[39]. Intriguingly, DEGs of PSCCE were significantly overrepresented in exclusion programs—of 146 repressed genes associated with CD8 cytotoxic T-cell exclusion, 52 were downregulated DEGs in PSCCE ($P < 1.0 \times 10^{-5}$, Monte Carlo stimulation of hypergeometric test). PSCCE and SCLC also had significantly higher OE scores of exclusion programs for both CD8 cytotoxic T cells and T cells (Fig. 6e).

## Discussion

In this study, we performed genomic, transcriptomic, and immune profiling on a rare but highly aggressive neuroendocrine malignancy arising from the alimentary tract–PSCCE. By integrating multi-omics data, we revealed the following findings: (1) PSCCE harbors a high frequency (98%) of *RB1* disruption mediated by multiple mechanisms; (2) the transcriptome of PSCCE highly resembles that of SCLC, but not that of ESCC or

EAC; (3) PSCCE has two distinct subtypes regulated by lineage TFs *ASCL1* and *NEUROD1*; and (4) insufficient T-cell infiltration is widely observed in PSCCE.

*RB1* was disrupted in up to 93% of all SCLC[12,42]. The previous WES profiling of PSCCE reported a *RB1* mutation frequency of 27%[11]. Here, we demonstrated that with an integrated analysis of WES, RNA-seq, and IHC, we could identify *RB1* disruption in 98% of PSCCEs with high fidelity. We also discovered that exon deletions—small deletions affecting as few as one exon—were a major and unique *RB1*-disrupting mechanism in PSCCE. It should be noted that none of WES, RNA-seq, or IHC alone succeeded to detect all *RB1* disruption events in PSCCE. Given that only two-thirds of *RB1* disruption events in the present study were detectable by the canonical WES, we proposed a potential underestimate of *RB1* disruption in previous WES studies of cancers. Nearly universal disruption of *RB1*, together with paucity of *CCND1* amplification or *CDKN2A* deletion, comprised genomic features distinguishing PSCCE from ESCC or EAC.

In the present study, unsupervised clustering of gene expression patterns revealed a molecular-based taxonomy of cancers different from the present organ-of-origin classification. We discovered that PSCCE had a transcriptome much more akin to SCLC. This shared transcriptomic landscape of PSCCE and SCLC was consistent to the neuroendocrine differentiation and loss of

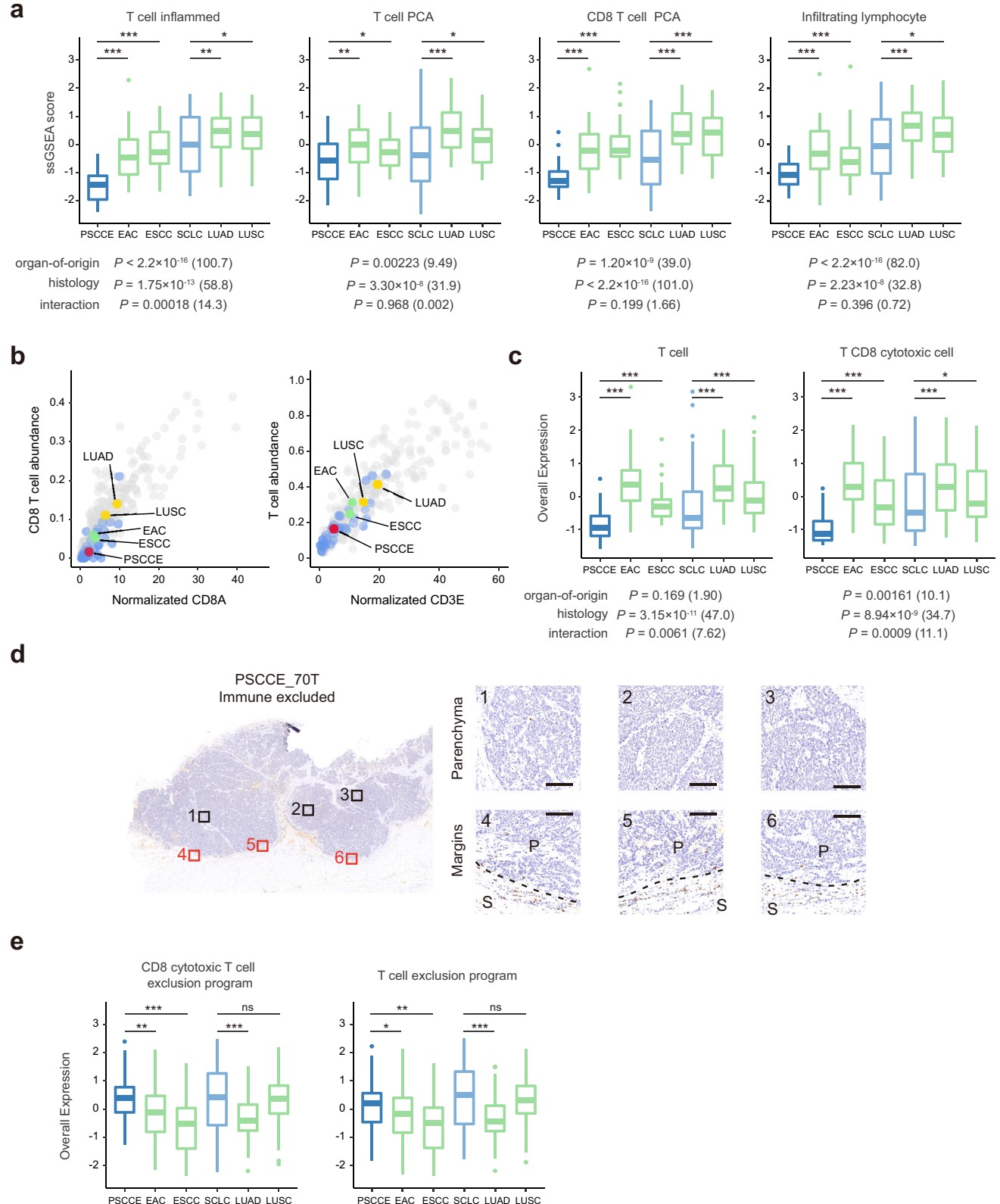

RB1 in both entities. On the other hand, transcriptome of PSCCE was quite different from ESCC or EAC.

Recent research of SCLC categorized it into four molecular subtypes[15]. Whether these subtypes also exist in other neuroendocrine cancers is relevant to oncogenesis mechanisms and

therapy vulnerabilities. Two subtypes of PSCCE, namely PSCCE-A and PSCCE-N, highly resembled the SCLC-A and SCLC-N subtypes of SCLC, in the aspects of transcriptomic features, regulatory factors, associated genomic alterations, and impact on patients' prognoses[13,15]. It is quite striking that molecular

**Fig. 6 PSCCE presented a T-cell excluded phenotype. a** ssGSEA scores of T cells and CD8 T cells related signatures in each cancer type. The impacts of "organ-of-origin" and "histology" on scores were determined using two-way ANOVA tests. Degree of freedom: organ-of-origin: 1; histology: 1. F values of each variate are shown in brackets behind P values. Source data are provided as a Source Data file. **b** T cell and CD8 T-cell abundance estimated by CIBERSORT are plotted against marker genes (CD3E and CD8A) expression level. PSCCEs are plotted in blue and other cancers in gray. The medians of each cancer type are plotted separately and annotated on the plot. **c** Overall Expression (OE) scores of T cells and CD8 cytotoxic T cells in each cancer. The impacts of "organ-of-origin" and "histology" on OE score were determined using the same method mentioned above. **d** Representative CD8A IHC staining of PSCCE. Left: overview of the fields distribution in whole section. Right top: fields of tumor parenchyma; right lower: fields of invasive margin (dashed line) between the parenchyma (P) and surrounding stroma (S). Scale bar: 100 μm. **e** OE scores of exclusion program for CD8 cytotoxic T cells and T cells in each cancer. In all panels, the upper bound, centerline, and lower bound of boxplot represent the 75 percentile, the median and the 25 percentile of data; the upper and lower whiskers extend to the largest and smallest value within 1.5 times of IQR from corresponding bound. Data beyond the whiskers are plotted as outlier dots. *$P < 0.05$, **$P < 0.01$, ***$P < 0.001$, ns not significant, by the Wilcoxon rank-sum test.

subtyping persisted in cancers arising from different organs and from patients with different ethnic backgrounds. Given the highly similar genetic and transcriptomic features of PSCCE and SCLC, it appeared intuitive that subtyping would also be shared. However, the molecular fundaments of *ASCL1*- and *NEUROD1*-regulated subtypes in both cancers remain largely unknown. It was once supposed that SCLC-N lesions might have a different cell-of-origin from SCLC-A lesions, due to the absence of *Neurod1*+ cells in mouse lungs[13]. Recently, a murine model of SCLC combining *Trp53* and *Rb1* knockout with *Myc* overexpression revealed that SCLC-N tumors could arise from SCLC-A precursor lesions[14], suggesting lineage plasticity fueled by *Myc*. Consistent with this idea, PSCCE-N tumors were also associated with *MYC* amplification.

Although we observed shared *ASCL1*- and *NEUROD1*-regulated subtypes, we did not confirm any *POU2F3*- or *YAP1*-regulated subtypes in PSCCE. One possible cause is our limited sample size. SCLC-P and SCLC-Y subtypes are relatively rare in SCLC[15]. If the proportions of subtypes were similar in PSCCE, a PSCCE-P or PSCCE-Y subtype could simply be missed due to rarity. Further studies are required to elucidate whether other molecular subtypes exist in PSCCE.

The similarities in terms of genomic alterations, transcriptomic features, and molecular subtypes between PSCCE and SCLC lead to another intriguing question: whether these two cancers have the same cell-of-origin, or some shared genomic alterations (such as loss of *TP53* and *RB1*) restricted lineage to small-cell carcinoma? Pulmonary neuroendocrine cells were reported to be the major cell-of-origins of SCLC[43]. A recent study found that neuroendocrine stem cell (NE$^{stem}$), which started uncontrolled proliferation upon loss of *TP53* and *RB1*, could be the cell-of-origin of SCLC[44]. However, evidence supporting such cells in the esophagus is lacking. Another emerging hypothesis is that certain genomic alteration enhances lineage plasticity and promotes cells to differentiate along a neuroendocrine trajectory. In support of this idea, *Trp53* and *Rb1* dual knockout was indispensable in all genetic-engineered murine models of SCLC[14,43,45]. Lung adenocarcinomas (LUAD) with loss of *RB1* were more likely to transform into SCLC after treatment[46,47], supporting the role of *RB1* loss in lineage switching. Our observations here emphasize role of *RB1* disruption in PSCCE. *RB1* loss also shaped the shared transcriptome landscape of PSCCE and SCLC. However, our study could not disclose the underlying mechanisms. Further research, including cell line and mouse model studies, is required.

Through the computational dissection of TME, we found that PSCCE had a TME characterized by insufficient infiltration of both T cells and CD8 cytotoxic T cells. IHC of PSCCE tumors revealed that a large fraction of PSCCE presented a CD8 T-cell "excluded" phenotype. CD8 T-cell "excluded" was also the predominant immune phenotype of SCLC, and was associated with failure of immune checkpoint blockage[48]. Multiple lines of

evidence indicated that the small-cell carcinoma histology was associated with insufficient T-cell infiltration. Gene expression programs associated with T-cell exclusion[39] significantly overlapped with DEGs of PSCCE and were highly expressed in PSCCE, indicating that a modulated transcriptional network in small-cell carcinoma might be responsible. Further research are required to provide more comprehensive description of various immune populations in PSCCE.

Our integrated study also provided therapeutic insights into PSCCE. Systemic chemotherapy regimens extrapolated from SCLC proved beneficial to PSCCE patients[1,49]. By providing rationales through molecular similarities, our observation supported the extrapolation of systemic treatment. The identification of subtypes of PSCCE, which were highly similar to SCLC-A and SCLC-N subtypes of SCLC, encouraged the extrapolation of future targeted therapy designed and trialed for certain SCLC subtypes into corresponding PSCCE subtype. The differences between PSCCE-A and PSCCE-N tumors also warranted molecular characterization when designing researches. Our profiling also introduced insight for future preclinical research for PSCCE. Highly and specifically expressed genes might be therapeutic targets, including *E2F* family[50] and *DLL3*[51]. Notch pathway reactivation may suppress essential neuroendocrine programs of PSCCE, leading to tumor regression[52]. The induction of T-cell inflamed TME by antibody-guided chemokines[53], or oncolytic viruses armed with recruiting chemokines[54] might improve response rate of immune therapy.

## Methods

**Study design and inclusion criteria.** This study was approved by the Ethics Committee of National Cancer Center/Cancer Hospital, Chinese Academy of Medical Sciences and Peking Union Medical College (2018103113071202). The study was conducted in accordance with local laws and the guidelines of Declaration of Helsinki. PSCCE patients were screened from patients who received radical esophagectomy for esophageal cancer in the Department of Thoracic Surgery, Cancer Hospital, Chinese Academy of Medical Sciences from 2006 to 2017. All patients provided written informed consent. Both FF and FFPE specimens of PSCCE tumors and matched esophageal tissue were obtained from institutional biobank. Pathological diagnoses were independently confirmed by two certificated pathologists (S. Shi and L.X.). Tumors with positive IHC staining for at least one of CD56, Synaptophysin, or Chromagranin A were included. Tumors were also assessed to have ≥60% tumor cell purity and without extensive necrosis. Matched non-cancerous esophageal samples were provided as normal controls in sequencing studies

**DNA and RNA extraction.** DNA from FF specimens was extracted using Allprep DNA/RNA/miRNA Universal Kit (QIAGEN) following the manufacturer's protocol. DNA and RNA from FFPE specimens were extracted using Allprep DNA/RNA FFPE Kit (QIAGEN) by WuXi NextCODE (Shanghai). The amount of DNA was determined by Qubit2.0 (Thermo Fisher) and integrity of DNA was determined by agarose electrophoresis. The amount of extracted RNA from FFPE sample was determined by NanoDrop (Thermo Fisher). Quality of RNA from FFPE sample was assessed by Agilent 2100 Bioanalyzer (Agilent) to have a $DV_{200}$ (percentage of RNA fragments >200 nucleotides fragment distribution value) ≥30%.

**Whole-exome sequencing (WES)**. WES library was prepared using the SureSelect Human All Exon Kit (Agilent, V5) following the manufacturer's protocol. Libraries were sequenced on Illumina NovaSeq platform, with the aim to obtain sequencing depth of 200× and 100× for tumor and normal tissue, respectively.

**Somatic mutation calling**. Sequenced reads were trimmed by Trimmomatic[55] (0.36) and mapped to reference genome hg19 by Burrows–Wheeler Aligner[56] (0.7.12-r1044). PCR duplicates were marked by Picard (https://broadinstitute.github.io/picard/, 2.9.0) and excluded from further analysis. Somatic mutations were detected using MuTect2[57] (3.5) with assistance from Genetron Health (Beijing). After integration of results and cross-validation of mutation calls, we further filtered mutation calls with following filters: (1) coverage at mutation site in tumor ≥10; (2) number of variant reads in tumor ≥3; (3) number of variant reads in matched normal sample ≤2; (4) maximal frequency in population (Exome Aggregation Consortium and 1000 Genome) <0.01; (5) mutations registered in snp142 database were removed unless it was also recorded in the COSMIC database (v70); (6) variant AF should be significantly higher in tumor than in matched normal tissue (examined by the Fisher's exact test with $P < 0.05$ and Benjamini–Hochberg corrected $q < 0.1$); (7) distribution of forward and backward reads in variant and reference reads covering mutation site should show no significant difference ("strand bias," examined by the Fisher's exact test with $P > 0.0001$).

**SMGs and genes with mutation cluster**. SMGs were called by MutsigCV[26] (1.4). Genes with significantly clustered mutations were identified using a nonrandom clustering method implanted in R package iPAC. A cluster ≤100 amino acids was considered within one functional domain and was included. Further, as mutations in genes that were not expressed conferred neutral impact in tumorigenesis, only gene that was expressed in tumors (defined by having expression ≥10.0 transcript per million (TPM) in ≥10% of 38 tumors with RNA-seq data) were considered as candidate driver genes. In analysis of combined cohort of 101 PSCCE, we only considered genes that were mutated in ≥5 cases (4.95% of combined cohort) in order to further identify genes of relevance.

**Validation of somatic mutations**. PCR primers were designed online using Primer3Plus (https://primer3plus.com/) to amplify mutation loci in *TP53*, *RB1*, and *NOTCH1*. As DNA fragmentation in FFPE samples greatly hampered amplification of long amplicons, we limited amplicons for FFPE DNA < 500 bp. A PCR failure was declared after three failed attempts. Sanger sequencing of PCR product was performed on 3730xl DNA Analyzer (Applied Biosystems). For loci that suffered PCR amplification failure, expression of somatic mutations in RNA-seq reads was checked by Integrative Genomics Viewer[58] (IGV, 2.6.2).

**Mutational signature analysis**. Mutational signatures were decomposed using a nonnegative matrix fraction method implemented in R package SomaticSignatures[59]. We included mutations from the present study and those reported by Wang et al.[11] into a "combined cohort" of PSCCE. SCLC mutations were also included to discover potential signatures unique to small-cell carcinomas. Optimal number of output signatures ($K$, candidate range from 2 to 20) was determined by 1000 iterations at each $K$ value. For each $K$, the cosine similarities between resultant signatures and COSMIC signatures (version 2) were calculated. Optimal $K$ was determined when (i) explained variance did not increase remarkably by further increasing $K$, and (ii) the average of cosine similarities to the most resembling COSMIC signatures reached maximum.

**SCNV calling**. SCNVs were called using CNVkit[60] (0.9.5). FF samples and FFPE samples were analyzed separately as two groups. Normal samples in each group were pooled up to serve as reference used in corresponding group. Coverages of each bin (each bin equals a capture region by WES) were compared against reference to calculate log2 ratios. Bins with log2 ratio < −15.0 were discarded. Segments were called from bins with same copy number. Segments were filtered for copy-number variations (CNVs) in healthy population. CNVs in healthy population were downloaded from Database of Genomic Variants[61] (DGV, http://dgv.tcag.ca/dgv/app/home, 2020-02-25 release). We built a common CNVs reference set (DGV.refset), containing CNVs with frequency >1% in studies with sample size ≥1000 included by DGV. Segments with a log2 ratio > 0.2 were filtered for gains in DGV.refset; segments with a log2 ratio < −0.2 were filtered for losses. Segments, which had >50% reciprocal overlap with DGV.refset CNVs, were excluded. Stringent thresholds were applied to call amplifications (log2 ratio > 0.807 which equaled >3.5 copies) and deletions (log2 ratio < −2.0 which equaled <0.5 copies). Significant SCNV peaks were called from DGV.refset filtered segments using GISTIC2.0[24] module (version 6.15.28) on GenePattern public server (https://cloud.genepattern.org/).

**Validation of SCNVs**. Copy number of *MYC*, *TERT*, and *SOX4* in all samples was determined using 7900HT real-time PCR system (Applied Biosystems). *IFNG* was used as the reference. When SCNVs were observed in IFNG locus, an alternative reference including *AQP5*, *ACACA*, and *ACLY* was used. SCNV results were

determined with same thresholds (amplifications: >3.5 copies; deletions: <0.5 copies) and were compared with SCNVs called from WES data.

**Validation of RB1 exon deletion breakpoints**. We designed series of PCR primers according to *RB1* exon deletions detected by SCNV analysis pipeline to gradually approach and finally locate exact breakpoints of deletions. As long PCR amplicons (>2000 bp) were required to efficiently amplify regions flanking breakpoints, validation was performed only on FF samples. Downstream breakpoint of PSCCE_15T was predicted to reside in large intergenic region and was not validated due to technical difficulty. Sanger sequencing of PCR products was performed by GENEWIZ (Suzhou) on 3730xl DNA Analyzer (Applied Biosystems).

**RNA sequencing (RNA-seq)**. Library was prepared using the TruSeq RNA Access Library Prep Kit (Illumina), which was optimized to provide reproducible result of RNA from FFPE samples. Paired-end 150-bp sequencing of the subsequent libraries was sequenced on Illumina NovaSeq platform, with the aim to obtain ≥6-Gb sequencing data.

**RNA-seq data analysis**. Sequencing reads were trimmed by Trimmomatic[55] (0.36) and aligned to reference genome hg19 by STAR[62] (2.4.2a). RSEM[63] (v1.3.1) was used to estimate abundance of annotated genes. Expression value in TPM was supplemented in Source Data 2.

DEGs were identified using R package DESeq2[64], by comparing 38 tumors against 23 matched normal esophageal samples sequenced together in the present study. "apeglm" parameter of DESeq2 was activated to accurately estimate true effect size. Genes with $P < 0.01$, $q < 0.05$, and log2FoldChange ≥ 1.0 by DESeq2 were defined as upregulated DEGs; genes with $P < 0.01$, $q < 0.05$, and log2FoldChange ≤ −1.0 were defined as downregulated DEGs.

Gene ontology and Kyoto Encyclopedia of Genes and Genomes enrichment was performed using R package clusterprofiler[65]. Enrichment with $P < 0.01$ and $q$ value < 0.05 was considered significant. Pathways that were significantly enriched for upregulated DEGs but not for downregulated DEGs were considered upregulated; similarly, pathways which were specifically enriched for downregulated DEGs were considered downregulated.

GSEA[66] was performed using "GSEA" module at GenePattern server (https://cloud.genepattern.org/).

**Identification of RB1 exon skipping events**. Sashimi plot of RNA-seq bam files was visualized and generated by IGV[58] (2.6.2). Sashimi plots of driver genes in PSCCE (*TP53*, *RB1*, and *NOTCH1*) were manually checked for reads spanning abnormal exon junctions. Abnormal exon junctions with ≥10 supporting reads were included in further analyses. Sashimi plots of 23 matched normal sample were used as control.

**Comparison with TCGA tumors and SCLC**. RNA-seq reads count data of TCGA esophageal cancers[21] were downloaded from the National Cancer Institute Genomic Data Commons data portal (https://portal.gdc.cancer.gov/projects/TCGA-ESCA) for DEG analysis. Gene expression data of 81 SCLC samples reported by George et al.[12] were obtained from Supplementary Table 10 of ref. [12]. Batch-effect-normalized TOIL recomputed TPM gene expression data of TCGA LUAD[31], LUSCs[30], and esophageal cancers[21] were downloaded from UCSC Xena data hubs (https://toil.xenahubs.net/download/tcga_RSEM_gene_tpm.gz).

**Multiple cancers clustering**. Thirty-eight EAC samples and 38 ESCC samples from TCGA were selected randomly to match PSCCE in number. Similarly, 81 LUAD samples and 81 LUSC samples from TCGA were matched to 81 SCLC samples reported by George et al.[12] (list of samples used in analysis in Supplementary Data 9).

Expression data of selected samples were combined and quantile-normalized to minimize the batch effect. Genes with high expression (average expression in top 50%) were included and then log2-transformed. Median absolute deviations (MAD) of genes were calculated. Top $G$ genes with highest MAD were median-centered and then supplied as input of consensus clustering using R package ConsensusClusterPlus[67]. Candidate number of resultant groups $N$ was set ranging from 2 to 10; candidate number of input genes $G$ was set 1000, 2000, 3000, 4000, and 5000. For each $N$ and $G$, 1000 times of fraction and clustering were iterated. The output tracking plots, cluster matrices heatmaps, and CDF curve were manually checked. $N$ was finally determined as four as: (i) change in CDF sharply decreased to nearly zero at 4; (ii) extremely small group ($n \leq 5$) representing outliers begin to emerge from $N \geq 4$; (iii) the three major groups (Group NE, Squamous, and Adeno) remained relatively stable when further tuning of $N$ and $G$. Due to its small size, Group 4 was not included for further analysis and not plotted in Fig. 3c. Genes associated with the clustering most were identified using R package ropls[68] and top 800 genes were plotted in Fig. 3c.

After optimal clustering was determined, comparisons of $G$ genes expression among $N$ groups were performed using the Tukey's HSD test. Genes with

log2Foldchange ≥ 2.0, $P < 0.05$, and Benjamini–Hochberg corrected $q < 0.1$ against other groups were considered signature genes and listed in Supplementary Data 10.

**Identification of subtypes in PSCCE.** Genes with high expression (average expression ≥5.0 TPM) were log2-transformed. MAD of genes were calculated. Top $G$ genes with highest MAD were median-centered and then supplied as input of consensus clustering by R package ConsensusClusterPlus[67]. Candidate number of clusters $N$ ranged from 2 to 10 and candidate numbers of input genes $G$ were 1000, 2000, 3000, 4000, and 5000. For each $N$ and $G$, 1000 times of fraction and clustering were iterated. The output track plot, cluster matrix heatmap, and CDF curve were manually checked. One stable group of nine samples (PSCCE-N) was repeatedly observed. The other 29 samples had unstable clustering but seldom clustered with PSCCE-N samples, thus were studied collectively as PSCCE-A. Genes associated with the clustering most were identified using R package ropls, and top 800 genes were plotted in Fig. 4a.

To better discover signature genes for each subtype, we compared all genes with an average expression ≥1.0 TPM between subtypes using the Tukey's HSD test. Genes with log$_2$Foldchange ≥ 2.0, $P < 0.05$, and Benjamini–Hochberg corrected $q < 0.1$ against other groups were considered signature genes and listed in Supplementary Data 12.

SCNVs that affected ≥3 of 38 RNA-sequenced tumors were tested for significantly different distribution between two subtypes using the Fisher's exact test. Chromosomal location enrichment analysis of SCNVs with $P < 0.05$ was performed on Enrichr (http://amp.pharm.mssm.edu/Enrichr/). Chromosomal locations with $P < 0.01$ and $q < 0.05$ were considered significant.

**Computational dissection of TME in PSCCE.** Infiltrating immune cell signatures were collected from literatures (Supplementary Data 14). Gene expression profiles of PSCCE, SCLC, LUAD, LUSC, EAC, and ESCC samples in Supplementary Data 9 were combined and quantile-normalized. ssGSEA scoring of the signatures was performed using R package GSVA. The deconvolution estimation of immune cells' abundance was performed using CIBERSORT[38] (https://cibersort.stanford.edu/). For CIBERSORT, SCLC data were not included due to missing value in several genes. scRNA-seq-derived cell type signatures, immune cell exclusion signatures, and OE scoring source code were obtained from report by Jerby-Arnon et al[39]. Raw scores of immune signatures are provided in Source Data 4. Median-centered and standard-deviation-scaled scores were used to generate boxplots in Fig. 6 for better visualization. Monte Carlo stimulation was carried out by repeating random draw of $L$ gene (repressed genes in CD8 cytotoxic T-cell exclusion signature) from 37536 gene (genes whose expression values were not all zero in 38 PSCCEs and 23 matched normal samples) for 100,000 times and recording $M$, the numbers of certain 2249 genes (downregulated DEGs in PSCCE) in each draw. Then, $P(M \geq m)$ equals (times when $M \geq m$)/100,000. Stimulation was repeated three times with different random seeds and the maximal $P$ value was reported.

**Immunohistochemistry (IHC).** To guarantee consistency of IHC procedure, staining of Rb, Ascl1, and Neurod1 was performed on tissue microarray sections of PSCCE tumors and matched normal esophageal samples. Briefly, cylindrical tissue cores of tumor and matched normal samples were extracted from paraffin blocks and re-embedded into a microarray block. Microarray block was sectioned. Sections were deparaffinized in xylene, rehydrated in serial ethanol solution and distilled water. Heat-mediated antigen retrieval was performed in pH9.0 Tris-EDTA solution. Sections were blocked and incubated with primary antibody and secondary antibody. Sections were developed with DAB and counterstained with hematoxylin.

**T-cell phenotype profiling by IHC.** Whole slides of tumor section covering both the bulk of tumor and surrounding normal tissue were stained for CD8A. Evaluation was carried out by certificated pathologists (S. Shi and L.X.). First, ten random high power fields (HPF, 400×) of tumor parenchyma >300 μm from outermost border of tumor were checked for CD8A-positive cells. Efforts were taken to evenly distribute the HPFs inside tumor parenchyma. If ≥10 CD8A-positive lymphocytes were observed to directly contact with tumor cell in ≥3 HPF (for tumors with large volume or extensive connective-tissue septa, 20 fields were checked and ≥6 HPF were then used as threshold) then immune phenotype of examined tumor was considered "inflamed." Otherwise, ten HPFs around the invasive margin were checked. If ≥10 CD8A-positive lymphocytes were observed in ≥3 HPF along the invasive margin, then immune phenotype was considered "excluded"; if ≥10 CD8A-positive lymphocytes were merely observed in <3 HPF, the examined tumor were considered "desert." Profiling results were listed in Supplementary Data 15.

**Antibodies and PCR primers.** Antibodies and PCR primers used in this study were listed in Supplementary Tables 1 and 2, respectively.

**Statistics.** The R program (3.6.1) was used for statistics. Differences between groups were examined by the Fisher's exact test or Wilcoxon rank-sum test where appropriate. All tests were two-sided unless stated otherwise. Survivals of different groups were compared using the log-rank test. HR and its 95% confidential interval were estimated using Cox proportional hazard model. Significant level was set to 0.05 unless stated otherwise.

**Reporting summary.** Further information on research design is available in the Nature Research Reporting Summary linked to this article.

## Data availability

Raw WES and RNA-seq data generated by this study has been deposited in the European Genome-Phenome Archive (EGA) under accession number EGAD00001007647, in the Genome Sequence Archive for human (GSA-human) under the accession number of HRA000488, and in the National Omics Data Encyclopedia (NODE) under the accession numbers OEP001067 and OEP001068. Processed WES and RNA-seq data has been deposited in European Genome-Phenome Archive (EGA) under accession number EGAS00001004889. The TCGA data used in this study are available at the National Cancer Institute Genomic Data Commons (GDC) data portal [https://portal.gdc.cancer.gov/projects/TCGA-ESCA]. Gene expression data of TCGA lung adenocarcinomas[31] (LUAD), lung squamous cell carcinomas[30] (LUSC), and esophageal cancers[21] used in this study are available at the UCSC Xena data hubs [https://toil.xenahubs.net/download/tcga_RSEM_gene_tpm.gz]. Source data are provided with this paper. The remaining data are available in the Article and Supplementary Information.

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

## Acknowledgements

This work was supported by the National Key R&D Program of China (2018YFC1312100 to J.H. and Y.G., 2017YFC1311000 to Y.G. and F.T., 2018YFC1313100 to Z.L., 2019YFC1315700 to R.L.), the National Natural Science Foundation of China (81972316 to Y.G.), Beijing Nova Program (Z181100006218032 to Y.G.), Beijing Hope Run Special Fund of Cancer Foundation of China (LC2017A14 to S. Shi), the CAMS Initiative for Innovative Medicine (CIFMS) (2017-I2M-1-005 to J.H., 2017-I2M-2-003 to F.Z., 2016-I2M-1-001 to S.G.), and Non-profit Central Research Institute Fund of Chinese Academy of Medical Sciences (2019PT320022 to Y.L.).

## Author contributions

Y.G. and J.H. jointly oversaw, coordinated, and provided funding for this study. Y.G. conceptualized and designed analyses and experiments. J.H. established the patient cohort. J.H., S.G., Q.X., F.T. and Y.L. participated in collection and biobanking of the specimens. S. Shi, L.X. and Y.W. performed pathological review of specimen and assessment of IHC stain. R.L., Z.Y., F.S., H.C., W.G. and F.Z. participated in extraction and quality control of nucleic acids. R.L. performed data analysis, with support from Y.G., Z.Y., S. Sun, Z.L., N.B., J.W., and Y.S. R.L., Z.Y., S. Sun, Z.L., and Y.G. participated in conceptual design and generation of plots and tables. The manuscript was written by R.L., edited by Y.G., and approved by all authors.

## Competing interests

The authors declare no competing interests.
