## [Peer Review File · Nature Communications]

Reviewers' comments:

Reviewer #1 (Remarks to the Author):

Review

Li and colleagues present a multi-omic analysis of small cell cancers of the esophagus. This descriptive study demonstrates that these tumors are more similar to small cancer of the lung than to EAC or ESCC—which is not a surprising conclusion yet still good to demonstrate. While this will ultimately make a useful contribution to the literature, some improvements are needed both in writing and in some of the analysis and discussion.

1. Of note, they contrast a prior paper with exomes of PSCCE—finding 80% p53 but only 27% RB1.... This is contrasted with their much higher assessment of RB1 loss- but from multiple ways of assessing—it would be useful to clearly compare the rates of somatic mutation with this prior paper.
2. There are some concerns regarding the quality of the copy number data. I have concerns that many of the most significant peaks are spurious—this may specifically be a problem with joint analysis of FFPE and fresh samples as copy-number is challenging with FFPE samples.... Are any peaks selective to the FFPE samples? There are not enough details on how these data are processed and what methods are used to properly clean the results. Many focal peaks are present but not discussed—the fear is that many are artifacts. Also, they are rather selective in describing the targets of the peaks—for example, they cite MYC as an amplified target but myc is outside the peak listed for 8q24.
3. Are the mutations in CREBBP and EP300 hotspots seen in any other cancers? If novel and largely in FFPE samples, some validation is necessary.
4. They claim homozygous Rb1 deletion in 8 cases. The GISTIC results show only a borderline significant event—if it was a homozygous event in ~20% of tumors I would expect a much more significant result. How is 'homozygous' deletion called?
5. In Figure 2a—how do you explain a homozygous deletion and a splicing abnormality in the same tumors? The authors need to evaluate a collection of 'normal' tissues some and ideally other cancer RNAseq data with their analytics to determine how often these splice forms are seen.
6. Negative IHC for Rb1 is a difficult marker—can they benchmark their results in small cell cancers vs. other tumors with a larger dataset? Also, these results need to be read in a blinded fashion (e.g. a pathologist needs to read them without knowing which are small cell vs other) and verified by 2nd blinded reader.
7. Line 251—the statement that the 3 groups showed differential 'dependence' on oncogenic pathways is not correct as stated—dependence needs to be shown functionally and cannot be determined from gene set analyses
8. 5a as a data slide on the notch pathway is a schematic but does not show real data to support this point.
9. In the methods- it states that both fresh and FFPE—in the text of the paper, this breakdown is necessary to detail in more depth in terms of the breakdown of types. Also, there needs to be evaluation for possible systematic biases in results of fresh vs. FFPE results.
10. The writing needs correction/editing for more proper English grammar.

Reviewer #2 (Remarks to the Author):

Li et al. perform a molecular characterization, at a substantially deeper genomic level, and for the first time at the transcriptomic level, of esophageal small cell carcinomas (PSCCE). These tumors present similarities to small cell lung carcinomas (SCLC), including ubiquitous loss of TP53 and RB1 loss, alterations in NOTCH family members and in epigenetic regulators, common molecular subtypes and strong immune suppression. These data are novel, informative, and of potentially high impact, suggesting that the molecular features described in SCLCs may be extendable to other neuroendocrine small cell carcinomas. The analysis appears to be well-performed, and the text requires only minor grammatical editing.

I have a few comments and suggestions regarding the manuscript.

Most importantly, the authors should clearly state the statistical tests used for each analysis in the text, and especially in the figures. As one example, please state the statistical tests used to compare the differentially altered signaling pathways within the different esophageal cancer histologic types (Figure 3b). It is also unclear what the comparator is in the DEG analysis of PSCCE (Figure 3a). Is it normal tissue or the tumors from the other esophageal cancer histologic types? Please address throughout.

I would suggest a brief discussion of the potential significance of the enrichment of mutational signature E1 in PSCCE, especially in light of different etiologic drivers from SCLC (less strongly associated with tobacco exposure). What does this say about oncogenesis?

It would be interesting to see if the expression of the transcription factors defining the other two SCLC subtypes, POU2F3 and YAP1, are differentially expressed among the PSCCE samples. Figure 4A seems to show some samples with low ASCL1 and NEUROD1 expression and it would be interesting to comment if these may express POU2F3 or YAP1. I am sure the authors have looked at this, and if the answer is no, this could just be very briefly stated.

In SCLC, NEUROD1 high tumors exhibit milder neuroendocrine features than ASCL1 high counterparts (while still maintaining expression). Is this observation reproduced in PSCCE?

I would suggest that the authors provide a supplementary table including the clinical demographic features of the PSCCE cases, dividing by molecular subtype (ASCL1/NEUROD1), and stating the p-values of the comparisons of each clinical characteristic by subtype group.

Does multivariate analysis suggest that PSCCE subtype and NOTCH mutations are independent

prognostic factors?

Charles Rudin

Reviewer #3 (Remarks to the Author):

Primary small cell carcinoma of the esophagus (PSCCE) is a rare tumor with poor prognosis. It is histologically similar to small cell lung cancer (SCLC), and different from the other esophageal tumors, esophageal squamous cell carcinoma (ESCC) and esophageal adenocarcinoma (EAC). Li et al. performed whole exome sequencing of 46 PSCCE and matched normal samples, RNA sequencing for 38 of the tumors and 25 matched samples, roughly doubling the number of profiled PSCCE in the public domain (previously 55 by Wang et al.). They identified mutations and copy number variation, and compared with SCLC and the two other types of esophageal cancer, EAC and ESCC. Similar to SCLC, alterations in RB1 were identified in almost all PSCCE tumors (45/46). In 41 tumors, RB1 protein was missing by IHC. They also identified two sub-types based on transcriptional profiling. The less frequent sub-type (9/38) was more coherent and stably clustered together. The subtypes correspond to two of the five subtypes of SCLC, which may have clinical implications. PSCCE tumors excluded T-cells.

Major comments:

1. The novel subtypes suggested should be shown on the previously published PSCCE cohort of Wang et al.
2. Clustering which puts samples from different datasets in different clusters is very likely to be due to the inherent difficulty in combining datasets. Quantile normalization does not remove all technical differences. This is not discussed at all. There are ways to convince the reader this is not merely a technical effect, e.g. that normal samples of the same tissue from different datasets cluster together, or that tumors of the same type from different datasets cluster together. The fact that the Wang PSCCE data is not included is also strange.
3. The author did not identify PSCCE subtypes corresponding to two subtypes of SCLC by unsupervised analysis. Are those the least frequent subtypes? It may be worth to try supervised analysis to see if the marker genes of those subtypes are expressed in some tumors.
4. Is the NOTCH status different between the subtypes?
5. Why were only T cells tested?
6. L524 requiring that a gene is expressed in at least half of the tumors to call it expressed for the purpose of candidate driver gene is unnecessarily too strict. A driver gene in 45% of tumors may still be extremely important. 10% is more reasonable, and possibly a higher expression level threshold, as driver genes are likely to be highly expressed in tumors where they are amplified.

Minor comments:

1. The term 'insufficient immune cell infiltration' may better be replaced by 'T cells exclusion', as insufficient have unjustified implications (insufficient for what?), and only T cells were tested. PSCCE may actually be enriched for other immune cell types.
2. In mutation identification analysis, it will be informative to add the total number of mutations, not just the nonsynonymous. Synonymous mutations might also have an effect, especially if identified in the same gene in multiple tumors. The reader is left to wonder if the 4713 in L113 is the total

number of mutations or just of single nucleotide substitutions and not indels, for example.

3. Similarly, in L167-169, the reader is left to wonder how many TP53 were in total, assuming 52 stands for all nonsynonymous mutations.

4. L 107-109 "The mutation rate of PSCCE in this study was significantly higher than that reported by Wang et al12 ($P = .0225$), owing to the higher sequencing depth." – What was Wang mutation rate? This claim should be supported by subsampling the reads to similar sequencing depth and repeating the analysis.

5. Several methods are not cited appropriately. No reference for MutsigCV, MuTect2, CNVkit, mutational signatures (L536), RSEM, DESEQ, Consensus clustering and more.

6. L162-169 described the genes identified as significantly mutated. The L170-176 list other genes that were mutated – how were those identified? If by manual inspection, how many genes were manually inspected? What was the selection criteria? How many of those were mutated?

7. L182 – homozygous deletion of a gene is by definition mutually exclusive with a mutation in that gene. Possibly partial deletion?

8. L223,262 – differential expression analysis of PSCCE against what? Matched normal samples? Of which tissue? Other cancer types?

9. L232 seed should be see

10. Figure 3c – labels seem to be misplaced.

11. Figure 3g.g – what is the order of the genes?

12. L278 and elsewhere – consider replacing changeable by unstable.

13. L289 'integrated analysis revealed' – what type of analysis? How significant is the association?

14. L293, 294 – what test and significance level were used?

15. L308 – consistent with which of the findings described before?

16. L322, 324 – significant by which test? P-value?

17. L334-335 – where is this finding shown?

18. Figure 5d – what do N and T stand for?

19. L357 Projection on what? The PSCCE RNA-seq data?

20. L362 Deconvolution does not confirm signature projection, merely repeat the analysis – those are two computational ways to show the same thing based on the same data.

21. L388 field should have probably be field

22. L479 ethnic should probably be ethic

23. L536-L539 – Consider replacing the last two sentences of the paragraph.

24. L545 – RNA sequencing description is not detailed enough – read length and paired or non paired reads should be specified.

25. L548 – what makes a sequence read validated? How many of the reads were not validated?

26. In several places (L594, L516, L609 and elsewhere), p-value threshold is stated as 0.05, and though multiple comparisons were done, no correction is mentioned. State type of correction if performed, or perform correction if not performed.

27. L608 what filtering was done prior to differential expression analysis?

We greatly appreciate the reviewers' time and constructive comments. To address the concerns of the reviewers, we performed additional analyses and validations, and revised our manuscript accordingly. We believe that the inspiring comments from the reviewers helped us substantially improve our manuscript.

Please find our responses below:

Reviewer #1 (Remarks to the Author):

Review

Li and colleagues present a multi-omic analysis of small cell cancers of the esophagus. This descriptive study demonstrates that these tumors are more similar to small cancer of the lung than to EAC or ESCC—which is not a surprising conclusion yet still good to demonstrate. While this will ultimately make a useful contribution to the literature, some improvements are needed both in writing and in some of the analysis and discussion.

1. Of note, they contrast a prior paper with exomes of PSCCE—finding 80% p53 but only 27% RB1.... This is contrasted with their much higher assessment of RB1 loss- but from multiple ways of assessing—it would be useful to clearly compare the rates of somatic mutation with this prior paper.

Response:

Thank you for pointing this out. Nineteen nonsynonymous somatic mutations of *RB1* affected 16 out of 46 cases (34.8%) in our study, comparable to previous report by Wang et al (27.3%, 15/55, $P = 0.5166$, Fisher's exact test).

We compared somatic mutations identified by WES and reported in the revised Manuscript as the following (newly added part is underlined):

“WES identified 19 somatic mutations of *RB1* in 16 PSCCE tumors. Mutation frequency of *RB1* was comparable to that reported by Wang et al (15/55, 27.3%, $P = 0.5166$, Fisher's exact test). Seventy-nine percent (15/19) of RB1 ...”

2. There are some concerns regarding the quality of the copy number data. I have concerns that many of the most significant peaks are spurious—this may specifically be a problem with joint analysis of FFPE and fresh samples as copy-number is challenging with FFPE samples.... Are any peaks selective to the FFPE samples? There are not enough details on how these data are processed and what methods are used to properly clean the results. Many focal peaks are present but not discussed—the fear is that many are artifacts. Also, they are rather selective in describing the targets of the peaks—for example, they cite MYC as an amplified target but *myc* is outside the peak listed for 8q24.

Response:

We appreciated very much for this comment. CNV calling from FFPE WES is challenging, due to non-uniform distribution of coverage and fragmentation of FFPE DNA (Ref 1).

Following reviewer's suggestion, we thoroughly checked and revised our CNV analysis pipeline,

data cleaning process and results. In the renewed CNV analysis pipeline, we cleaned our results mainly by the following measures:

- (1) PCR duplicates were marked with Picard (<https://broadinstitute.github.io/picard/>, 2.9.0) and removed from further analysis. This step was also performed but not mentioned in the initial Manuscript. This information was now added to method section.
- (2) Two piled-up references, namely **fresh-frozen reference** and **FFPE reference**, were built from fresh-frozen normal samples and FFPE normal samples, respectively. Piled-up reference guarded against random fluctuation introduced during library preparation and sequencing process. Piled-up FFPE reference would also help to reduce non-uniform distribution of sequencing coverage in normal samples.
- (3) CNV analysis was performed **separately** for fresh-frozen and FFPE samples. Specifically, fresh-frozen tumors were compared against fresh-frozen reference; FFPE tumors were compared against FFPE reference.
- (4) Bins (each bin equals a single capture region of WES) whose log₂ coverage ratio against reference < -15.0 against reference were highly likely due to poor capture or alignment (Ref 2). These segments were excluded from further analysis.
- (5) As high coverage depth (> 300×) was achieved in present study, a justified resolution to one bin could be achieved (Ref 2). We allowed report of segments containing as few as one bin (compared to ≥ 5 bins in old version results) to decipher small SCNVs.
- (6) Segments were filtered for CNVs that were commonly observed in healthy population. We built a reference set of CNVs in healthy population (DGV.refset) from Database of Genomic Variants data (Ref 3, <http://dgv.tcag.ca/dgv/app/home>, hg19 assembly, 2020-02-25 release). Briefly, we included common CNVs (defined by population frequency > 1%) from large-scale (defined by ≥ 1000 participants) profiling studies to represent true population frequency. Ten studies, including 1000 Genome Consortium Phase 1 and Phase 3 studies, were included, amounting to 17647 CNVs from 62139 healthy participants. Then segments called from PSCCE tumors were compared against DGV.refset. Segments with a > 50% reciprocal overlap with CNVs from DGV.refset were considered commonly observed variations in healthy human and excluded from further analysis.
- (7) The default thresholds (also the thresholds used in initial Manuscript) for calling a SCNv in CNVkit were relatively loose, mainly to guarantee sensitivity in low purity tumor samples. We applied a stringent threshold of amplifications (log₂ ratio > 0.807, equals > 3.5 copies) and deletions (log₂ ratio < -2.0, equals < 0.5 copies) to discover most prominent variations, which were high confident true amplifications and deletions.

With the aforementioned measures applied, we observed the following feature in the revised CNV results:

- a) Segments generated by revised pipeline moderately resembled those from initial pipeline. A median of 52.0% (range: 21.9%-79.7%) revised segments had ≥ 80% reciprocal overlap

with segments generated from initial pipeline. Numbers of segments showed no significant difference between revised results and initial results, both in fresh-frozen samples (median: 284 versus 315, $P = 0.1272$, paired Wilcoxon test) and FFPE samples (median: 277 versus 276, $P = 0.6811$, paired Wilcoxon test)

- b) Some “spurious” peaks in Fig 1d and Supplementary Table S3 were no longer identified as significant by GISTIC because corresponding segments were identified as CNVs commonly observed in healthy population and discarded. Table R1 showed some representative peaks that were discarded. Filtering with DGV.refset discarded a median of 0.16% (range: 0.04% - 0.58%) of all bins. Although few in amount, it seemed to contribute much to the changes of GISTIC plot.

Table R1 Peaks in initial GISTIC results that were discarded due to filtering

Peak location	Type	Genes in this peak	Population frequency	Overlapping CNV accession ID*
17q21	Gain	NSF, ARL17A	2.8%-3.4%	dgv3254n100, nsv516807
17q21	Loss	NSF, ARL17A	5.9%-14.5%	dgv3254n100, esv3893017
14q11	Gain	OR4K5, OR4K1, OR4M1	3.6%-24.4%	dgv1766n100, dgv33e203
15p11	Loss	OR4N4, OR4M2	2.3%-17.9%	dgv2373n100, esv2659516
8p23	Loss	DEFB105A, DEFB106A	1.6%-7.9%	esv2761169, dgv6936n100
1q21	Loss	PPIAL4A, PPIAL4B	19.6%-98.8%	dgv4e203, esv3587491

*Multiple overlapping CNVs may be identified. Only two representative accession IDs are listed.

- c) We discovered more *RB1* “exon deletions” by the revised SCN calling pipeline. We found homozygous deletion events often affected only some exons (median: 11 exons, range: 1 to 25) but not full gene body of *RB1*. We use “exon deletions” to replace the imprecise “homozygous deletions of *RB1*” used in initial Manuscript. Table R2 listed all 13 *RB1* exon deletions and 1 homozygous deletion of whole *RB1* locus in PSCCE. The median length of *RB1* exon deletions was 52.5kb (range: 0.2kb to 2828kb). The revised *RB1* exon deletions were highly consistent with splicing-abnormalities observed in RNA-seq (Fig R1 and Supplementary Fig 2 in revised Manuscript). We designed PCR primers to locate exact breakpoint of *RB1* exon deletions in PSCCE_10T, PSCCE_12T, PSCCE_13T, PSCCE_14T and PSCCE_16T, which was highly consistent with SCN results (Fig R1, and Fig 2, Supplementary Fig 2 and Supplementary Table 7 in revised Manuscript). Validation of breakpoints in FFPE samples were severely hampered by DNA fragmentation.

Table R2 Fourteen deletions event affecting *RB1* gene

#	Sample	Genomic Ranges	Length/kb	Affected exons
1	PSCCE_10T	chr13:48916724-48955589	38.9	Exon 3 to exon 17
2	PSCCE_12T	chr13:49030345-49030538	0.2	Exon 19
3	PSCCE_13T	chr13:48916724-48955589	38.9	Exon 3 to exon 17
4	PSCCE_14T	chr13:49047412-49047640	0.2	Exon 24

5	PSCCE_15T	chr13:48985562-49109906	124.3	Exon 18 to exon 27 of RB1 Plus part of downstream RCBTB2 gene
6	PSCCE_16T	chr13:48934140-48986599	52.5	Exon 7 to exon 17
7	PSCCE_29T	chr13:49027124-49054249	27.1	Exon 18 to exon 27
8	PSCCE_31T	chr13:48956089-49089882	133.8	Exon 18 to exon 27 of RB1 Plus part of downstream RCBTB2 gene
9	PSCCE_40T	chr13:48934140-49051603	117.5	Exon 7 to exon 26
10	PSCCE_47T	chr13:49027124-51855425	2828.3	Exon 3 to exon 27 of RB1 Plus downstream numerous genes
11	PSCCE_61T	chr13:48916724-48985062	68.3	Exon 3 to exon 17
12	PSCCE_65T	chr13:48934140-49106586	172.4	Exon 7 to exon 27 of RB1 Plus part of downstream RCBTB2 gene
13	PSCCE_77T	chr13:49037870-49039526	1.7	Exon 21 to exon 23
14	PSCCE_79T	chr13:48807471-50881863	2074.4	Full length of RB1 Plus downstream numerous genes

*Rows with green background are newly identified exon deletions by our revised SCNv calling pipeline.

Fig R1 Representative *RB1* exon deletions and resultant splicing abnormalities in PSCCE. Schematic plot of exon deletions and validation of genomic breakpoints, sashimi plot of *RB1* mRNA and Rb protein immunohistochemistry of (a) PSCCE_12T, PSCCE_14T and PSCCE_16T; (b) PSCCE_10T which suffered

different deletions on each allele of *RB1*. In schematic plot of exon deletions, grey dashed lines denotes deleted region, black rectangles denote *RB1* exons and green ones denote *ITM2B* exons. Curves in sashimi plot represent reads spanning each exon junctions with number of reads denoted. Abnormal junctions are plotted in red. Matched normal samples are provided as positive control for IHC. Scale bar: 50 microns.

- d) We validated copy-number of 102 loci in 34 samples (12 fresh-frozen and 22 FFPE samples, *MYC*, *TERT* and *SOX4* for each sample) by quantitative PCR. Thirty five of 36 (97%) loci in fresh-frozen samples and 61 of 66 (92%) loci in FFPE samples showed consistent results between WES and quantitative PCR.

We followed the reviewer's comment to search selective peaks in FFPE samples. Numerous GISTIC peaks appeared selective to FFPE samples (Fig R2). For example, peak 8q24 harboring *MYC* were called from FFPE group but not observed in fresh-frozen group. However, *MYC* was also amplified in 3 fresh-frozen tumors. We propose SCNVs at 8q24 in the relatively small fresh-frozen group (n = 13) might fail to get statistically significant simply due to limited observation numbers. Similarly, 13q14 peak harboring *RB1* deletion were not called in fresh-frozen group, but *RB1* deletions were observed in 6 fresh-frozen tumors. Amplification peak on 6p22 was selective to FFPE samples. Genes in this peak, including *E2F3* and *SOX4*, were amplified in none of 13 fresh-frozen samples. However, the difference was not statistically significant (for example, *SOX4* were amplified in 0/13 of fresh-frozen sample versus 4/33 of FFPE sample, $P = 0.313$ by Fisher's exact test). We also validated *SOX4* amplification in PSCCE_36T and PSCCE_40T by qPCR (Supplementary Table 4 in revised Manuscript).

Fig R2 GISTIC q-plot of significant SCN V peaks observed in (a) thirteen fresh-frozen samples; (b) thirty three FFPE samples; and (c) all 46 samples.

In summary, we fully revised our SCN V analysis pipeline. By applying multi-step data cleaning and

more stringent thresholds, we obtained a revised SCNv result considerably improved than the previous one. We performed validation of SCNvs called from WES data, including (1) *RB1* deletions by directly capturing breakpoint and comparing mRNA abnormalities with deletions and (2) *MYC*, *TERT*, *SOX4* amplification by quantitative PCR. These validations together consolidated our bioinformatics analytic results.

Following reviewer's suggestion, we added one paragraph to the Method section to describe our SCNv analysis pipeline, data cleaning and threshold we used in our revised Manuscript, as the following:

"Somatic copy number variations (SCNVs) were called using CNVkit⁵⁸ (0.9.5). Fresh-frozen samples and FFPE samples were analyzed separately as two groups. Normal samples in each group were pooled up to serve as reference used in corresponding group. Coverages of each bin (each bin equals a capture region by WES) were compared against reference to calculate log₂ ratios. Bins with extremely low log₂ ratio (< -15.0) were discarded. Segments were called from bins with same copy number. Segments were filtered for copy number variations (CNVs) in healthy population. CNVs in healthy population were downloaded from Database of Genomic Variants⁵⁹ (DGV, <http://dgv.tcag.ca/dgv/app/home>, 2020-02-25 release). We built a common CNVs reference set (DGV.refset), containing CNVs with frequency > 1% in studies with sample size ≥1000 included by DGV. Segments with a log₂ ratio > 0.2 were filtered for gains in DGV.refset; segments with a log₂ ratio < -0.2 were filtered for losses. Segments which had > 50% reciprocal overlap with DGV.refset CNVs were excluded. Stringent thresholds were applied to call amplifications (log₂ ratio > 0.807 which equaled > 3.5 copies) and deletions (log₂ ratio < -2.0 which equaled < 0.5 copies). Significant SCNv peaks were called from filtered segments files using GISTIC 2.0 module (version 6.15.28) on GenePattern public server (<https://cloud.genepattern.org/>)."

Ref 1: Robbe P et al. *Genet Med*. 2018;20(10):1196-1205.

Ref 2: The CNVkit manual. <https://cnvkit.readthedocs.io/en/stable/pipeline.html>

Ref 3: MacDonald JR et al. *Nucleic Acids Res*. 2014;42:D986-992.

3. Are the mutations in CREBBP and EP300 hotspots seen in any other cancers? If novel and largely in FFPE samples, some validation is necessary.

Response:

Yes, all the SNVs of *EP300* and *CREBBP* occurred in sites which have been described in numerous previous works using fresh-frozen tumor samples.

Among them, Y1414 of *EP300* is a hotspot which have been reported in our previous report of esophageal cancers (Ref 1), as well as in SCLCs (Ref 2), TCGA bladder cancers (Ref 3) and liver cancers (Ref 4). D1435 and W1502 of *CREBBP* are hotspots, which have been reported in multiple hematopoietic and lymphoid malignancies (Ref 5-7) and bladder cancers (Ref 8).

Indels affecting *EP300* (n = 1) and *CREBBP* (n = 1) have not been previously described (COSMIC database v91, searched at 2020-05-06). We validated them using Sanger sequencing (Fig R3).

Fig R3 Sanger sequencing validation of indels in EP300 and CREBBP. (a) Sanger sequencing results in PSCCE_70T and matched normal sample. (b) Sanger sequencing results in PSCCE_8T and matched normal sample. Red dash line denote deletion of two bases in PSCCE_8T.

Reference in this response:

Ref 1: Gao et al. *Nat Genet.* 2014;46(10):1097-1102.

Ref 2: George et al. *Nature.* 2015;524(7563):47-53.

Ref 3: Robertson et al. *Cell.* 2017;171(3):540-556.

Ref 4: Kan et al. *Genome Res.* 2013;23(9), 1422-1433.

Ref 5: Pasqualucci et al. *Nature.* 2011;471(7337):189-195.

Ref 6: Zeithofer et al. *Leukemia.* 2012;26(8):1797-1803.

Ref 7: Morin et al. *Clin Cancer Res.* 2016;22(9):2290-2300.

Ref 8: Guo et al. *Nat Genet.* 2013;45(12):1459-1463.

4. They claim homozygous Rb1 deletion in 8 cases. The GISTIC results show only a borderline significant event—if it was a homozygous event in ~20% of tumors I would expect a much more significant result. How is ‘homozygous’ deletion called?

Response:

As described in response to comment 2, a segment with a log2 ratio against normal sample reference < -2.0 (equals to < 0.5 copy in a diploid genome) was considered “homozygous deletion”. Genes whose partial or all were involved in a deleted segment were then reported as “homozygous deletion”.

RB1 exon deletions in PSCCE were short in length (median: 52.5kb, range: 0.2kb to 2828kb; described in detail in response to comment 2). These short exon deletions might have been partially masked by improper noisy-reducing parameter (joint segment size) setting in initial Manuscript.

We re-set “joint segment size” to its default value of zero to include small SCNVs. With the revised SCNV results and updated GISTIC parameter, fourteen *RB1* deletions lead to a sharp 13q14 peak with q value of 6.91×10^{-9} (Fig R2 in response to comment 2).

In the revised Manuscript, we updated the parameters in GISTIC analysis and updated results presented in Fig 1d and Supplementary Table 3.

5. In Figure 2a—how do you explain a homozygous deletion and a splicing abnormality in the same tumors? The authors need to evaluate a collection of ‘normal’ tissues some and ideally other cancer RNAseq data with their analytics to determine how often these splice forms are

seen.

Response:

RB1 has an extraordinarily large gene body, with small exons separated by large introns. As was described in response to comment 2, most of the deletions observed in *RB1* gene deleted only part of but not whole *RB1* locus (“exon deletions”). The remaining exons of *RB1* could still be transcribed into mRNA and joint together to produce splicing abnormalities observed in RNA-seq. Fig R4 gives two examples of exon deletions as genomic basis for splicing abnormality.

Fig R4 Representative *RB1* exon deletions and splicing abnormalities in PSCCE. Schematic plot of exon deletions and validation of genomic breakpoint (left), sashimi plot of *RB1* mRNA and validation of abnormal exon junction (middle) and Rb protein immunohistochemistry (right) of (a) PSCCE_12T and (b) PSCCE_16T. Curves in sashimi plot represent reads spanning each exon junctions with number of reads denoted. Abnormal junctions were plotted in red. Matched normal samples were provided as positive control for IHC. Scale bar: 50 microns.

We used RNA-seq data from 23 matched normal esophageal samples sequenced together as normal control in our initial Manuscript. Here, following reviewer’s advice, we provided Sashimi plots of 23 matched normal esophageal tissue sequenced in our study (Fig R5). Additionally, we provided sashimi plots of 10 normal lung sample RNA-sequenced by our lab (unpublished data, Fig R6).

As was shown in Fig R5 and Fig R6, all normal samples showed a typical splicing pattern of *RB1* transcript ENST00000267163, which encoded full length Rb1 protein. No splicing abnormalities described in PSCCE tumors were observed in normal samples inspected. Two normal samples, PSCCE_13N and LUNG461, showed 13 reads and 12 reads supporting abnormal junctions connecting an *RB1* exon and some part of an intron (red curve in plot). However, abnormal junctions in PSCCE_13N and LUNG461 were not jointed to any upstream exons and were rare compared to normal reads.

Thus we confidently conclude that the splicing abnormalities described in PSCCE tumors were cancer-specific events.

Fig R5 Sashimi plots of *RB1* mRNA in 23 match normal esophagus samples. Curves represented reads spanning exon junctions, with number of reads denoted on the curve. Abnormal junction in PSCCE_13N was plotted in red.

Fig R6 Sashimi plots of *Rb1* mRNA in 10 normal lung samples. Curves represented reads spanning exon junctions, with number of reads denoted on the curve. Abnormal junction in LUNG461 was plotted in red.

6. Negative IHC for Rb1 is a difficult marker—can they benchmark their results in small cell cancers vs. other tumors with a larger dataset? Also, these results need to be read in a blinded fashion (e.g. a pathologist needs to read them without knowing which are small cell vs other) and verified by 2nd blinded reader.

Response:

Negative IHC marker bears the risk of producing false negative when IHC procedure fails. In awareness of this, the IHC of Rb1 showed as part of results in initial Manuscript was performed on a tissue microarray (TMA), with matched normal esophageal tissues alongside the corresponding tumor samples (Fig R7). By checking positive Rb1 signal in nuclei of normal samples, we could confidently exclude an IHC failure.

Fig R7 Representative overview and magnified fields of Rb1 IHC staining of PSCCE. Middle panel: Scanned picture of Rb1 IHC of PSCCE tissue microarray. Cores from matched normal esophagus were located in the same row to the right side of corresponding tumor cores. Magnified fields (200×) from PSCCE_13 and PSCCE_66 were shown in the left and right panel, respectively. PSCCE_13T showed positive Rb1 staining and was discussed in the initial Manuscript.

We also benchmarked the Rb1 IHC procedure in an additional cohort (benchmarking cohort) including 78 esophageal squamous cell carcinoma (ESCC), 17 large cell neuroendocrine carcinoma (LCNEC) and 47 SCLC. Rb1 protein IHC staining was performed in TMA, as was shown in Fig R8, by which we could use matched normal tissue as positive control. Two experienced pathologists, Dr. Susheng Shi and Dr. Liyan Xue performed scoring independently. Histology of samples and name of protein stained were concealed.

Fig R8 Representative snapshot of Rb1 IHC staining in TMA. Rb1 IHC staining was benchmarked in ESCC (a), LCNEC (b) and SCLC (c). IHC staining was performed on tissue microarray (TMA) to maximize processing consistency. Normal tissue on the same TMA could serve as positive control.

Results of Rb1 IHC in benchmarking cohort were summarized in Table R3 below. The proportion of Rb1 negative SCLC was highly similar to PSCCE. The proportion of Rb1 negative tumors in PSCCE were substantially higher than ESCC and LCNEC.

Table R3 Summary of Rb1 IHC staining in benchmarking cohort

Cancer type*	Total number of tumors	Rb1 negative tumors	Rb1 positive tumors	Reported RB1 alteration rate**
ESCC	78	30 (38%)	48 (62%)	7%-9% (Ref 1-2)
LCNEC	17	11 (65%)	6 (35%)	42% (Ref 3)
SCLC	47	46 (98%)	1 (2%)	93% (Ref 4)

*ESCC: esophageal squamous cell carcinoma; LCNEC: large-cell neuroendocrine carcinoma; SCLC: small cell lung cancer. **Somatic mutations and homozygous deletions were included.

Moreover, the proportion of Rb1 negative SCLC was highly comparable to *RB1* alteration rate revealed by WGS performed by George et al (Ref 4). The proportion of Rb1 negative tumors in ESCC and LCNEC were higher than the reported *RB1* alteration rate revealed by WES (Ref 1-3). These observations were consistent with our results that WES might fail to discover some *RB1* disrupting

events.

Based on the above evidences, we could confidently conclude that our Rb1 IHC evaluation pipeline is trustworthy. We added the Rb1 IHC of matched normal samples in revised Fig 2 and Supplementary Fig 2 as positive control. We also added a paragraph describing process of IHC staining in Method section of revised Manuscript.

Ref 1: Gao YB et al. *Nat Genet.* 2014;46(10):1097-1102.

Ref 2: Cancer Genome Atlas Research Network. *Nature.* 2017;541(7636):169-175.

Ref 3: George J et al. *Nat Comms.* 2018;9(1):1048.

Ref 4: George J et al. *Nature.* 2015;524(7563):47-53.

7. Line 251—the statement that the 3 groups showed differential ‘dependence’ on oncogenic pathways is not correct as stated—dependence needs to be shown functionally and cannot be determined from gene set analyses

Response:

Thank you for pointing this out. We replace L251 in original Manuscript with the following “The three groups also showed differential activation of oncogenic signaling pathways”.

8. 5a as a data slide on the notch pathway is a schematic but does not show real data to support this point.

Response:

We appreciate this comment from Reviewer 1. To better present dysregulation of Notch pathway gene expression, we showed frequency of upregulation and downregulation of each gene in 38 PSCCE tumors. The revised Fig 5a is like the following Fig R9.

Fig R9 Summary of dysregulation of Notch pathway gene expression. Color intensity denotes frequency of gene upregulation (Up, defined as having an expression value > 3 times of 95% quantiles of expression value of 23 matched normal samples) or downregulation (Dn, defined as having an expression value < 1/3 expression of 5% quantile of normal samples) in PSCCE tumors. Somatic mutation (Mut) rate of NOTCH 1-3 were also shown.

9. In the methods- it states that both fresh and FFPE—in the text of the paper, this breakdown is necessary to detail in more depth in terms of the breakdown of types. Also, there needs to be evaluation for possible systematic biases in results of fresh vs. FFPE results.

Response:

Following reviewer's suggestion, we evaluated differences between fresh-frozen and FFPE DNA results in the following aspects:

- (1) The proportion of C>T in FFPE samples (53.1%) was higher than that in fresh-frozen samples (43.1%), consistent with previous report that FFPE samples tended to have higher C>T substitution due to deamination during fixation (Ref 1). C>T substitution was the most common SNV in both fresh-frozen and FFPE samples.
- (2) FFPE samples had higher mutational burden (overall mutation rate: 3.36/Mb and 2.28/Mb for FFPE and fresh-frozen samples respectively, $P = 0.00628$; nonsynonymous mutation rate: 2.54/Mb and 1.72/Mb for FFPE and fresh-frozen samples respectively, $P = 0.0044$, all by Wilcoxon rank-sum test). The proportion of low allele-frequency mutations (allele frequency < 0.05) in FFPE samples (23%, 887/3881) was significantly higher than that in fresh-frozen samples (16%, 164/1037, $P = 1.14 \times 10^{-6}$, Chi-square test). These observations were consistent with previous reports that FFPE samples harbored more mutations due to low allele frequency nucleotide transitions induced during fixation (Ref 2-3). Mutational burden of PSCCE, no matter compared collectively or separately, was still lower than ESCC, EAC and SCLC, ranking moderately among cancers sequenced by TCGA project.
- (3) We validated somatic mutations in *TP53* ($n = 54$), *RB1* ($n = 19$) and *NOTCH1* ($n = 15$). PCR amplification success rates were satisfactory for both fresh-frozen (100%, 24/24) and FFPE (86%, 55/64) samples. In loci successfully amplified, 24 of 24 (100%) in fresh-frozen sample and 56 of 56 (100%) in FFPE samples were validated by Sanger sequencing, supporting high fidelity of our somatic mutation results. We further validated 7 of 9 loci that suffered PCR amplification failure by confirming expression of somatic mutations in RNA-seq reads (Supplementary Fig 2 in the revised Manuscript).
- (4) As described in response to comment 2, we found by performing data cleaning and applying stringent threshold to pick out most prominent amplification and deletions. SCNv results were highly consistent with qPCR validation in both fresh-frozen and FFPE samples. "Selective" peaks in FFPE samples were mainly due to larger sample size but not systemic bias.

In summary, we did find certain differences between FFPE sample and fresh-frozen sample, all had been previously reported. Some influence was introduced by FFPE samples, such as mutational burden and enrichment of C>T substitution in mutational signature analysis. However, as discussed above, these influences were mild.

Sequencing and subsequent analysis of FFPE samples are challenging. For a rare cancer such as PSCCE, archival FFPE samples were probably the only choice for any sizable sequencing study. FFPE

samples were proved to have good performance in discovering driver mutations and actionable events (Ref 1-3). We also found that by stringent filtering and proper interpretation, FFPE WES sequencing provided trustworthy and valuable information on molecular characteristics of PSCCE.

In the revised Manuscript, we discussed the above aspects in corresponding sections. We also provided sample type information in Supplementary Table 1 for readers to comparing parameters of interest between two sample types.

Ref 1: Astolfi A et al. *BMC Genomics*. 2015;16:892.

Ref 2: Van Allen EM et al. *Nat Med*. 2014;20(6):682-688.

Ref 3: Robbe P et al. *Genet Med*. 2018;20(10):1196-1205.

10. The writing needs correction/editing for more proper English grammar.

Response:

We appreciate this comment from the Reviewer. Our revised Manuscript has been edited by American Journal Experts for more proper English grammar.

Reviewer #2 (Remarks to the Author):

Li et al. perform a molecular characterization, at a substantially deeper genomic level, and for the first time at the transcriptomic level, of esophageal small cell carcinomas (PSCCE). These tumors present similarities to small cell lung carcinomas (SCLC), including ubiquitous loss of TP53 and RB1 loss, alterations in NOTCH family members and in epigenetic regulators, common molecular subtypes and strong immune suppression. These data are novel, informative, and of potentially high impact, suggesting that the molecular features described in SCLCs may be extendable to other neuroendocrine small cell carcinomas. The analysis appears to be well-performed, and the text requires only minor grammatical editing.

I have a few comments and suggestions regarding the manuscript.

Most importantly, the authors should clearly state the statistical tests used for each analysis in the text, and especially in the figures. As one example, please state the statistical tests used to compare the differentially altered signaling pathways within the different esophageal cancer histologic types (Figure 3b). It is also unclear what the comparator is in the DEG analysis of PSCCE (Figure 3a). Is it normal tissue or the tumors from the other esophageal cancer histologic types? Please address throughout.

Response:

We appreciated your comment. We clarified the statistical tests by stating the following in the revised Manuscript (newly added part is underlined):

- (1) By enrichment analysis, we discovered several pathways that were largely distinguishable between PSCCE and esophageal cancers (Fig 3b). (In corresponding Method section) Gene ontology (GO) and Kyoto Encyclopedia of Genes and Genomes (KEGG) enrichment was performed using R package "clusterprofiler". Enrichment with $p < 0.01$ and Benjamini-Hochberg corrected q value < 0.05 were considered significant. Pathways that was significantly enriched for upregulated DEGs but not for downregulated DEGs were

considered upregulated; similarly, pathways which were specifically enriched for downregulated DEGs were considered downregulated.

- (2) We compared the gene expressions of 38 tumors against 23 matched normal esophageal samples to identify differentially expressed genes..... (In corresponding Method section) Differentially expressed genes (DEGs) were identified using R package “DESeq2”⁶², by comparing 38 tumors against 23 matched normal esophageal samples sequenced together in the present study.
- (3) The nonsynonymous mutation rate of PSCCE was significantly lower than those of ESCC (3.15/Mb, $P = 0.00021$), EAC (5.16/Mb, $P = 1.11 \times 10^{-7}$) and SCLC (8.62/Mb, $P < 2.2 \times 10^{-16}$, all by Wilcoxon rank-sum test)
- (4) *CCND1* amplification...and in only 4% of PSCCE ($P = 1.06 \times 10^{-7}$, Fisher’s exact test).
- (5) *CKDN2A* deletion...was only observed in 4% of PSCCE ($P = 2.46 \times 10^{-6}$, Fisher’s exact test).
- (6) *ERBB2* was amplified in...none of PSCCEs ($P = 7.36 \times 10^{-5}$, Fisher’s exact test).
- (7) The PSCCE-N subtype had a significantly higher *MYCN* level than PSCCE-A ($P = 0.0012$, Wilcoxon rank-sum test, Supplementary Fig 4b).

I would suggest a brief discussion of the potential significance of the enrichment of mutational signature E1 in PSCCE, especially in light of different etiologic drivers from SCLC (less strongly associated with tobacco exposure). What does this say about oncogenesis?

Response:

Signature E1 is highly similar to COSMIC mutational Signature 1 (v2, 2015), whose etiology is spontaneous deamination of 5-methylcytosine. We propose that the enrichment of signature E1 may provide clue of oncogenesis in the following aspects:

- (1) Endogenous and spontaneous deamination of 5-methylcytosine is the major mutagenesis mechanism in PSCCE.
- (2) The differences of tobacco-related signature may reflect an organ-specific vulnerability to particular mutagens. The esophagus was not directly exposed to tobacco. Although smoking has been defined as risk factor for both ESCC (Ref 1-2) and EAC (Ref 2), the proportions of smoking related signature in ESCC and EAC are also low (Ref 3-4).

Ref 1: Lin Y et al. *J Epidemiol.* 2013;23(4):233-242.

Ref 2: Pennathur A et al. *Lancet.* 2013;381(9864):400-412.

Ref 3: Gao YB et al. *Nat Genet.* 2014;46(10):1097-1102.

Ref 4: Secrier M et al. *Nat Genet.* 2016;48(10):1131-1141.

It would be interesting to see if the expression of the transcription factors defining the other two SCLC subtypes, POU2F3 and YAP1, are differentially expressed among the PSCCE samples. Figure 4A seems to show some samples with low ASCL1 and NEUROD1 expression and it would be interesting to comment if these may express POU2F3 or YAP1. I am sure the authors have looked at this, and if the answer is no, this could just be very briefly stated.

Response:

We appreciate this comment from Dr. Rudin. Whether a *POU2F3* or *YAP1*-dominated subtype existed in PSCCE was also of our concern.

We identified 5 sample lower for both *ASCL1* and *NEUROD1* (both < 10.0 TPM), which were listed in Table R4. *POU2F3* in these five tumors were also very low.

Table R4 Expression of four transcriptional regulator in *ASCL1*^{low}*NEUROD1*^{low} PSCCEs

Sample	ASCL1	NEUROD1	POU2F3	YAP1
PSCCE_6T	1.64	4.43	1.36	11.41
PSCCE_13T	1.11	1.99	1.57	9.52
PSCCE_75T	0.05	5.40	0.11	16.66
PSCCE_63T	6.11	0.44	0.12	7.41
PSCCE_21T	5.63	8.91	0.80	38.99

* All in unit of Transcript Per Million (TPM)

YAP1 expressions values were slightly higher. We found that it might be due to a higher organ-specific baseline of *YAP1*. We compared *YAP1* and *POU2F3* expression to *ASCL1* and *NEUROD1*, and compared the ratios with SCLC-P and SCLC-Y samples. SCLC-Y subtype had both high *YAP1/ASCL1* ratios and *YAP1/NEUROD1* ratios (Fig R10). However, we did not observe any PSCCE had such outstanding expression of *YAP1* (that is, 38 PSCCEs in the present study had relatively high level of either *ASCL1* or *NEUROD1*). Moreover, *YAP1*-dominated subtype was reported to be associated with intact Rb protein, while Rb was completely abolished in all 38 RNA-sequenced tumors. We propose that the evidences obtained from the present study was insufficient to claim a *YAP1*-regulated subtype.

Fig R10 Comparisons of regulatory transcription factor expression levels. (a) *POU2F3/NEUROD1* ratios are plotted against *POU2F3/ASCL1* ratios. PSCCE samples were compared with reported SCLC-P samples. Subtypes are plotted in different colors according to color legend. Red arrowhead denotes one sample with infinite *POU2F3/ASCL1* ratio. (b) *YAP1/NEUROD1* ratios are plotted against *YAP1/ASCL1* ratios. PSCCE samples were compared with reported SCLC-Y samples. Subtypes are plotted in different colors according to color legend. Red arrowheads denote two sample with infinite *YAP1/ASCL1* ratio. Expression values of SCLC-P and SCLC-Y tumors

were collected from Ref 3 according to Ref 1. Expression values of SCLC-P and SCLC-Y cell line were downloaded from Cancer Cell Line Encyclopedia data portal (<https://portals.broadinstitute.org/ccle/>) according to Ref 2 and Ref 4

We clarified our observation and reasoning about *POU2F3* and *YAP1*-dominated subtypes in the Results section of revised Manuscript, by stating:

“*POU2F3* and *YAP1* levels in the 38 RNA-sequenced PSCCEs were relatively low compared to *ASCL1* and *NEUROD1* and did not appear to be a selective master regulator (Supplementary Fig 5). Given the *ASCL1*- or *NEUROD1*-regulated expression patterns observed in our 38 RNA-sequenced PSCCEs, the evidence obtained in the present study was insufficient to confirm a *POU2F3*- or *YAP1*-dominated subtype.”

We also discussed the potential cause of the paucity of *POU2F3*- or *YAP1*-dominated subtype in the Discussion section, by stating:

“Although we observed shared *ASCL1*- and *NEUROD1*-regulated subtypes, we did not confirm any *POU2F3*- or *YAP1*-regulated subtypes in PSCCE. One possible cause is our limited sample size. SCLC-P and SCLC-Y subtypes are relatively rare in SCLC. If the proportions of subtypes were similar in PSCCE, a PSCCE-P or PSCCE-Y subtype could simply be missed due to rarity. Moreover, our inclusion criteria required expression of NE markers while SCLC-Y and SCLC-P subtypes were reported to be low for NE markers. Further studies are required to elucidate whether other molecular subtypes exist in PSCCE.”

Ref 1: Rudin CM et al. *Nat Rev Cancer*. 2019;19(5):289-297.

Ref 2: McColl K et al. *Oncotarget*. 2017;8(43):73745-73756.

Ref 3: George J et al. *Nature*. 2015;524(7563):47-53.

Ref 4: Huang YH et al. *Genes Dev*. 2018;32(13-14):915-928.

In SCLC, NEUROD1 high tumors exhibit milder neuroendocrine features than ASCL1 high counterparts (while still maintaining expression). Is this observation reproduced in PSCCE?

Response:

Yes, this observation was reproduced in PSCCE.

We observed that PSCCE-N subtype had significantly lower level of neuroendocrine marker *DDC* and *GPR* than PSCCE-A subtype (Fig R11a, Wilcoxon rank-sum test, *P* value shown in plot), consistent with reports by Carney et al (Ref 1) and Gazdar et al (Ref 2). The significantly different levels of *DDC* and *GRP* were shown as part of Fig 4b in initial Manuscript.

In terms of neuroendocrine markers widely used for diagnosis of both SCLC and PSCCE (Fig R11b), we found PSCCE-A subtype had significantly higher level of *NCAM1* (also known as CD56), while *CHGA* (Chromogranin A) and *SYP* (synaptophysin) were comparable between two subtypes.

Fig R11 Expression value of neuroendocrine markers. (a) Expression level of genes whose expression was reported to be distinct between two types of SCLC cell lines. (b) Expression level of neuroendocrine markers used for diagnosis of SCLC and PSCCE in clinical practice. Difference examined by Wilcoxon's rank-sum test.

Ref 1: Carney D et al. *Cancer Res.* 1985;45(6):2913-2923.

Ref 2: Gazdar A et al. *Cancer Res.* 1985;45(6):2924-2930.

I would suggest that the authors provide a supplementary table including the clinical demographic features of the PSCCE cases, dividing by molecular subtype (ASCL1/NEUROD1), and stating the p-values of the comparisons of each clinical characteristic by subtype group.

Response:

We appreciate this suggestion. We provide a new Supplementary table 13 same with Table R5 below, comparing clinical characteristics between two subtypes.

Major clinical characteristics, including age of diagnosis, gender, smoking, alcohol drinking, TNM stages and adjuvant therapies, showed no significant difference between two subtypes of PSCCE.

Table R5 Comparison of clinical characteristics of each subtype

	PSCCE-A (n = 29)	PSCCE-N (n = 9)	P value*
Age**	59.86±10.77	62.44±7.40	0.470
Gender			
Female	9	2	1
Male	20	7	
Smoking			0.5316
No	12	8	
Yes	17	7	
Drinking			0.1212
Never	16	2	
Occasional	5	1	
Regular (daily)	8	6	
Lesion site			0.5881
Upper thoracic	5	1	
Middle thoracic	16	4	
Lower thoracic	8	4	

pT stage			0.7004
T1	10	2	
T2	7	3	
T3	9	4	
T4	3	0	
pN stage			0.8434
N0	7	3	
N1	10	2	
N2	11	4	
N3	1	0	
TNM stage (AJCC 8th)			0.935
I	4	1	
II	7	2	
III	15	6	
IVA	3	0	
Adjuvant chemotherapy			0.6391
Yes	15	6	
No	6	1	
NA***	8	2	
Adjuvant radiotherapy			0.6729
Yes	6	3	
No	12	4	
NA***	11	2	

* Age was compared using Wilcoxon rank-sum test; others were compared using Fisher's exact test.

**Age: mean \pm standard deviation.

***NA: not available.

Does multivariate analysis suggest that PSCCE subtype and NOTCH mutations are independent prognostic factors?

Response:

No, neither molecular subtype nor NOTCH status was independent prognostic factor by multivariate analysis in our cohort of 38 patients with RNA-seq data.

In univariate analysis (Table R6), we discovered that male gender, smoking, AJCC TNM 8th stage IV, no adjuvant chemotherapy and subtype of PSCCE-N were significantly associated with poorer prognosis in 38 PSCCE with RNA-seq data. *NOTCH1-4* somatic mutation (SNVs and indels) was no longer associated with overall survival in RNA-seq cohort, perhaps due to a further limited sample size.

Table R6 univariate analysis of prognostic factor for PSCCE patients

Variates	Number	HR (95% CI)*
Age above 60		$P = 0.552$
No	18	Ref

Yes	20	1.265 (0.5834-2.741)
Gender		
		P = 0.0439
Female	11	Ref
Male	27	3.022 (1.031-8.861)
Smoking		
		P = 0.0101
No	14	Ref
Yes	24	3.218 (1.321-7.838)
Drinking		
		P = 0.245
Others	24	Ref
Regular	14	1.584 (0.7296-3.439)
pT stage		
		P = 0.0232
T1-3	35	Ref
T4a	3	4.424 (1.225-15.97)
pN stage		
		P = 0.633
N0	10	Ref
N1-3	28	1.269 (0.4774-3.373)
AJCC 8th TNM stage		
		P = 0.000289
I+II+III	35	Ref
IV	3	14.7 (3.436-62.85)
Adjuvant chemotherapy		
		P = 0.00634
No	7	Ref
Yes	21	0.1837 (0.05441-0.6201)
NA	10	
Adjuvant radiotherapy		
		P = 0.108
No	16	Ref
Yes	9	0.4483 (0.1685-1.193)
NA	13	
NOTCH1-4 status		
		P = 0.657
Wild type	27	Ref
Mutated	11	0.8209 (0.3435-1.962)
MYC amplification		
		P = 0.226
No	28	Ref
Yes	10	1.684 (0.7245-3.914)
Molecular subtype		
		P = 0.0394
PSCCE-A	29	Ref
PSCCE-N	9	2.437 (1.044-5.687)

*HR: hazard ratio; CI: confident interval

Multivariate analysis (Table R7) using Cox proportional hazard model identified adjuvant

chemotherapy as the only independent prognostic factor. TNM stage IV had undetermined HR, due to very limited samples size (n = 3).

Table R7 Multivariate analysis of prognostic factors for PSCCE patients

Variates	Hazard ratio (95% CI)	P value
Smoking: Yes	1.946 (0.5138-6.0528)	0.25
Adjuvant chemotherapy: Yes	0.1945 (0.0447-0.8459)	0.029
TNM stage (AJCC 8 th): IVA	NA	1
Molecular subtype: PSCCE-N	2.551 (0.8167-7.9655)	0.107

Based on results on hand, neither PSCCE subtype nor *NOTCH* status was independent prognostic factor of overall survival. However, we could not simply rule them out. Our sample size is too small to guarantee statistical power for multivariate analysis of 4 factors (empirically, 10 outcomes are recommended for each factor of interest, Ref 1). We hope future studies of PSCCE will provide more cases with molecular characterization and provide sufficient evidences to draw convincing conclusions.

Ref 1: Concato et al. *Journal of Clinical Epidemiology*. 1995;48(12):1495-1501.

Reviewer #3 (Remarks to the Author):

Primary small cell carcinoma of the esophagus (PSCCE) is a rare tumor with poor prognosis. It is histologically similar to small cell lung cancer (SCLC), and different from the other esophageal tumors, esophageal squamous cell carcinoma (ESCC) and esophageal adenocarcinoma (EAC). Li et al. performed whole exome sequencing of 46 PSCCE and matched normal samples, RNA sequencing for 38 of the tumors and 25 matched samples, roughly doubling the number of profiled PSCCE in the public domain (previously 55 by Wang et al.). They identified mutations and copy number variation, and compared with SCLC and the two other types of esophageal cancer, EAC and ESCC. Similar to SCLC, alterations in RB1 were identified in almost all PSCCE tumors (45/46). In 41 tumors, RB1 protein was missing by IHC. They also identified two sub-types based on transcriptional profiling. The less frequent sub-type (9/38) was more coherent and stably clustered together. The subtypes correspond to two of the five subtypes of SCLC, which may have clinical implications. PSCCE tumors excluded T-cells.

Major comments:

1. The novel subtypes suggested should be shown on the previously published PSCCE cohort of Wang et al.

Response:

In the paper by Wang et al., fifty-five PSCCE were sequenced by WES. However, no transcriptomic profiling, such as RNA-seq or gene expression microarray, was performed. It was technically impossible for us to integrate samples reported by Wang et al. into subtyping analysis, or to infer subtypes of samples from Wang et al. cohort.

The Wang et al. paper is freely available at <https://www.nature.com/articles/s41422-018-0039-1>.

2. Clustering which puts samples from different datasets in different clusters is very likely to be due to the inherent difficulty in combining datasets. Quantile normalization does not remove all technical differences. This is not discussed at all. There are ways to convince the reader this is not merely a technical effect, e.g. that normal samples of the same tissue from different datasets cluster together, or that tumors of the same type from different datasets cluster together. The fact that the Wang PSCCE data is not included is also strange.

Response:

We agree with the Reviewer. Cross-study comparison and clustering of RNA-seq data are challenging. Technical differences could be introduced in nearly every step of RNA sequencing, including RNA extraction, library preparation, sequencing platform and post-sequencing analysis (Ref 1-2). Many of technical differences are technically difficult to eliminate. We also agree that quantile normalization could not eliminate batch effects.

Each study included in the clustering only focus on one cancer type, thus we did not have same cancers from different datasets. Gene expression profiling was not performed by Wang et al. It was technically impossible for us to include Wang et al. PSCCE samples.

Following the Reviewer's suggestion, we evaluated remaining technical effect after quantile normalization by clustering of normal samples. We also compare clustering and downstream analyses results from quantile normalization with mainstream batch-effect correction tools, *ComBat* (Ref 3) and *BUScorrect* (Ref 4).

We collected matched normal esophageal samples in our cohort (n=23), normal lung samples in TCGA-LUAD (n=59), normal lung samples in TCGA-LUSC (n=50) and matched normal samples in TCGA-ESCA (n=13), yielding a raw gene expression matrix with 18523 genes and 502 samples (hereinafter referred to as "matrix").

When preparing gene expression matrix, we found huge heterogeneity in "normal esophageal samples". We discovered that the majority of normal samples in TCGA esophageal study (Ref 5) were actually **gastric mucosa**, as they highly express gastric marker progastricsin *PGC* but lack squamous cell markers *KRT14* and *TP63* (Table R8). Thus, in an ideally batch-effect-free scenario, only a **small fraction of TCGA-ESCA normal samples would cluster with our normal samples**.

Table R8 Expression value (TPM) of marker genes in normal samples and inferred tissue type

Samples	KRT14	TP63	PGC	ACTA2	Inferred tissue type
TCGA-L5-A43C-11	0	0.05	99347.77	24.48	Gastric mucosa
TCGA-V5-A7RE-11	12.02	1.07	1186.09	22.46	Gastric mucosa
TCGA-L5-A4OG-11	0.31	0.37	67150.01	147.78	Gastric mucosa
TCGA-L5-A4OJ-11	10.72	6.51	58489.9	72.23	Gastric mucosa
TCGA-IC-A6RE-11	293.98	58.84	0.06	251.17	Squamous mucosa
TCGA-IC-A6RF-11	8838.59	130.52	0.05	488.87	Squamous mucosa
TCGA-V5-AASX-11	0.13	1.54	6420.63	39.94	Gastric mucosa
TCGA-L5-A4OO-11	0.02	0.06	51162.53	1383.88	Gastric mucosa

TCGA-L5-A40F-11	22.14	1.4	9124.94	64.66	Gastric mucosa
TCGA-L5-A40M-11	244.53	11.5	7840.34	364.48	Gastric mucosa
TCGA-L5-A40R-11	5.57	0.22	54614.63	25.79	Gastric mucosa
TCGA-IG-A3I8-11	1.59	0.34	0	6875.27	Smooth muscle
TCGA-L5-A40Q-11	0	0.1	2.26	3765.43	Smooth muscle

We corrected expression matrix using quantile normalization, batch-effect correction algorithm *ComBat* and *BUScorrect*, respectively. We evaluate differences between datasets in each corrected matrix by dimension reduction using uniform manifold approximation and projection (UMAP, Ref 6). In brief, when performing batch-effect correction, each study was considered a batch. Gene with average expression in the upper 50% (n=9262) of corrected expression matrix were kept and log2 transformed. Then gene-wise median absolute deviation (MAD) were calculated. Top 2000 genes with highest MAD were median-centered and supplied to dimension reduction. The results were shown in Fig R12.

Fig R12 UMAP dimension reduction of (a) raw matrix and matrices corrected by (b) quantile normalization, (c) *ComBat* and (d) *BUScorrect*. Each point represented a sample. Samples with different histology were plotted in different color according to legend on the right.

As is shown in Fig R12a, raw expression matrix showed strong batch effect, as samples from each study tightly clustered together and no mixing of esophageal normal samples was observed. Quantile normalized (Fig RX12b), *ComBat* (Fig R12c) and *BUScorrect* (Fig R12d) all reduced batch effect, by clustering normal esophageal samples from TCGA (red dots) and the present study (light blue dots) together. In all scenario, most TCGA ESCA study normal samples (those suspected as gastric) clustered with esophageal adenocarcinoma, perhaps by their shared glandular features.

Then we compared downstream analyses of each corrected matrix. As is shown in Fig R13a, three principal groups were reproduced using all three correction methods. We compared identity of

each sample and found **98.3% (351/357)** and **98.0% (350/357)** samples pertained their group identity when using *ComBat* and *BUScorrect* correction methods, respectively. All 118 sample categorized into Group NE still clustered together as Group NE when using *ComBat* and *BUScorrect* correction. Both *ComBat* and *BUScorrect* corrected matrices produced similar heatmaps (Fig R13b) with three main groups corresponding to Group NE, Group Adeno and Group Squamous in the Manuscript.

Fig R13 Three stable groups were observed from gene expression matrices corrected by different method. (a) Track plot of consensus clustering using corrected matrices by three correction method. Each thin column represent one sample and color represented group identify. (b) Resultant heatmap of corrected gene expression matrices by *ComBat* and *BUScorrect*.

We also performed gene set enrichment analysis (GSEA) corrected matrices. We found the differential enrichment of pathways which we mentioned in the Manuscript, could also be reproduced from expression matrices corrected by *ComBat* and *BUScorrect* (Fig R14).

Fig R14 GSEA of RB1_loss signature in matrices corrected by (a) quantile normalization; (b) *ComBat*; and (c) *BUScorrect*. Similar selective activation of RB1_loss signature in Group NE against Group Adeno and Group Squamous (Group Ad&Sq) were observed.

Moreover, some results reported in the Manuscript also support that difference observed among three groups were not merely technical differences. In Fig 3e, group NE showed significantly higher *NCAM1*, *SYP* and *CHGA* level. These three genes were used as markers for SCLC and PSCCE in pathological diagnosis. Fig 3g showed a signature associated with *RB1*-deletion (Ref 7) was highly active in group NE, consistent with genomic profiling (which was less sensitive to technical effect) findings that *RB1* gene was disrupted universally in both PSCCE and SCLC but only in small fraction of Group Adeno and Group Squamous tumors.

In summary, we found that (a) quantile normalization did reduce batch-effect (but not fully eliminate for sure); (b) three widely used batch-effect minimization method produced highly similar results; (c) conclusions from quantile normalization could be cross-validated with genomic or pathological features not affected by RNA-seq batch-effect. As was shown and discussed above, we are confident that clustering **results shown in Fig 3d was not merely technical differences.**

If required, we would be happy to provide the full response to comment 2 as a “not merely technical effect” evidence in a Supplementary Note file or Peer Review file.

Ref 1: Leek JT et al. *Nat Rev Genet.* 2010;11(10):733-739.

Ref 2: Goh WWB et al. *Trends Biotechnol.* 2017;35(6):498-507.

Ref 3: Johnson WE et al. *Biostatistics.* 2007;8(1):118-127.

Ref 4: Luo X et al. *Journals of the American Statistical Association.* 2019;114(526):581-594.

Ref 5: Cancer Genome Atlas Research Network. *Nature*. 2017;541(7636):169-175.

Ref 6: McInnes L et al. ArXiv e-prints. 2018; 1802.03426.

Ref 7: Ertel A et al. *Cell Cycle*. 2010;9(20):4153-4163.

3. The author did not identify PSCCE subtypes corresponding to two subtypes of SCLC by unsupervised analysis. Are those the least frequent subtypes? It may be worth to try supervised analysis to see if the marker genes of those subtypes are expressed in some tumors.

Response:

Yes, the other two subtypes, SCLC-P and SCLC-Y, were less frequently observed. SCLC-P subtype was estimated to comprise 12%-18% of all SCLC (Ref 1). SCLC-Y subtype was estimated to account for 13% (7/51) of SCLC cell lines (Ref 2) and only 2.5% (2/81) of human primary SCLC (Ref 1).

We inspected expression of master regulators *POU2F3* and *YAP1*. When compared to SCLC-P (Ref 1, 4) or SCLC-Y (Ref 1-2) samples, *POU2F3* and *YAP1* expression in the studied PSCCEs appeared relatively low (Fig R15). SCLC-P samples had both high *POU2F3/ASCL1* ratios and high *POU2F3/NEUROD1* ratios. Similarly, SCLC-Y samples had both high *YAP1/ASCL1* ratios and high *YAP1/NEUROD1* ratios. No tumors high for both ratios were observed in present study; the studied PSCCEs were relatively high for either *ASCL1* or *NEUROD1*.

We added a paragraph discussing expressions of *POU2F3* and *YAP1* to the results section, concluding “*POU2F3* and *YAP1* levels in the 38 RNA-sequenced PSCCEs were relatively low compared to *ASCL1* and *NEUROD1* and did not appear to be a selective master regulator...the evidence obtained in the present study was insufficient to confirm a *POU2F3*- or *YAP1*-dominated subtype”. We also provided Fig R15 as a new Supplementary Fig 5 in the revised Manuscript.

Fig R15 Comparisons of regulatory transcription factor expression levels. (a) *POU2F3/NEUROD1* ratios are plotted against *POU2F3/ASCL1* ratios. PSCCE samples were compared with reported SCLC-P samples. Subtypes are plotted in different colors according to color legend. Red arrowhead denotes one sample with infinite *POU2F3/ASCL1* ratio. (b) *YAP1/NEUROD1* ratios are plotted against *YAP1/ASCL1* ratios. PSCCE samples were compared with reported SCLC-Y samples. Subtypes are plotted in different colors according to color legend. Red arrowheads denote two sample with infinite *YAP1/ASCL1* ratio. Expression values of SCLC-P and SCLC-Y tumors were collected from Ref 3 according to Ref 1. Expression values of SCLC-P and SCLC-Y cell line were downloaded from Cancer Cell Line Encyclopedia data portal (<https://portals.broadinstitute.org/ccle/>) according to

Ref 2 and Ref 4

Ref 1: Rudin CM et al. *Nat Rev Cancer*. 2019;19(5):289-297.

Ref 2: McColl K et al. *Oncotarget*. 2017;8(43):73745-73756.

Ref 3: George J et al. *Nature*. 2015;524(7563):47-53.

Ref 4: Huang YH et al. *Genes Dev*. 2018;32(13-14):915-928.

4. Is the NOTCH status different between the subtypes?

Response:

No, *Notch* status showed no significant difference between subtypes.

Two of 9 PSCCE-N tumors and 9 of 29 PSCCE-A tumors carried somatic mutations of *NOTCH1-4*. The mutation frequency showed no significant difference (2/9 versus 9/29, $P = 1$, Fisher's exact test).

We also checked expression level of Notch receptors (*NOTCH1-4*, Fig R16a), main effector (*HES1*), inhibitory effector (*HES6*) and inhibitory ligands (*DLL3* and *DLK1*, Fig R16b).

PSCCE-N subtype had significantly lower *DLL3* level than PSCCE-A subtype ($P = 0.00924$, Wilcoxon rank-sum test), as was denoted on heatmap of Fig 4a in Manuscript. However, *DLL3* level in both subtypes were significantly higher than normal esophagus samples (PSCCE-N versus normal samples, $P = 1.894 \times 10^{-6}$; PSCCE-A versus normal samples, $P = 8.466 \times 10^{-10}$, all by Wilcoxon rank-sum test), indicating a shared *DLL3* overexpression to suppress activity of *Notch* pathway.

Fig R16 Expression of genes in Notch pathway. Expression levels of (a) Notch receptors and (b) Notch effectors and ligands were compared among PSCCE-N, PSCCE-A and normal esophagus, using Wilcoxon's rank-sum test. * $P < 0.05$; ** $P < 0.01$; *** $P < 0.001$. Non-significant P values were shown on plots.

5. Why were only T cells tested?

Response:

We collected 19 signatures from literatures for ssGSEA scoring and 16 scRNA-seq derived signatures from Jerby-Arnon et al. (Ref 1) for OE scoring, including signatures for T cells, CD8 T cells, macrophages, NK cells, B cells, mast cells, neutrophils and eosinophils.

We observed multiple lines of evidence showing insufficient T cell infiltration in both PSCCE and

SCLC, as were shown in initial Manuscript. Some signatures reported in initial Manuscript were signatures for lymphocytes (including both B cells and T cells, “Module 4” and “Module 5” in initial Supplementary Fig 5a).

We chose to focus on T cells as: (1) low T cells and CD8 T cells abundance in PSCCE and SCLC were recurrently observed using signatures from multiple literatures; (2) T cells are considered major anti-tumor effector cells (Ref 2); (3) abundance and distribution of T cells and CD8 T cells are used as classifier for immune phenotyping (Ref 3) and predictors of immune therapy (Ref 4); (4) are studied by a plenty of research, which we could utilize to cross-validate each other and draw convincing conclusions.

Besides T cells, we also observed insufficient infiltration of macrophages (Fig R17a), which was another commonly observed population in TME. Some populations, such as B cells, appear to show inconsistent results in some analyses (Fig R17b). For the conciseness and preciseness of the Manuscript, we did not discuss them thoroughly.

In the revised Manuscript, we clarified that multiple immune populations were checked. We listed all source of signatures in revised Supplementary Table 14. We provided the raw scores for all immune signatures in a new Source Data 4 to give readers clues on immune infiltrates not described in main text. We also discussed limitation of our computational method in Discussion section to remind readers of need for future studies.

Fig R17 Comparisons of ssGSEA and OE scores for macrophages and B cells. Boxplots of ssGSEA and OE scores of macrophage-related signatures were shown in panel (a) and those of B-cell-related signatures in panel (b). Impact of organ-of-origin and histology was examined using two-way ANOVA (degree of freedom: Organ-of-origin = 1; histology = 1), with P value and F value shown below corresponding boxplot. Scores between

each pair of cancers was compared using Wilcoxon's rank-sum test. * $P < 0.05$; ** $P < 0.01$; *** $P < 0.001$.

Nonsignificant P values were shown on plot.

Ref 1: Jerby-Arnon L et al. *Cell*. 2018;175(4):984-997.

Ref 2: Appay V et al. *Nat Med*. 2008;14(6):623-628.

Ref 3: Mariathasan S et al. *Nature*. 2018;554(7693):544-548.

Ref 4: Ott PA et al., 2018. *Journal of Clinical Oncology*, 37:318-327.

6. L524 requiring that a gene is expressed in at least half of the tumors to call it expressed for the purpose of candidate driver gene is unnecessarily too strict. A driver gene in 45% of tumors may still be extremely important. 10% is more reasonable, and possibly a higher expression level threshold, as driver genes are likely to be highly expressed in tumors where they are amplified.

Response:

Following the Reviewer's suggestion, we re-defined "expressed" as "having expression levels ≥ 10.0 TPM in at least 10% of 38 tumors with RNA-seq data". Two candidate genes, *DNAH9* and *ODF1* were filtered by the old expression filter. Their expression levels still not satisfied the new expression filter of " ≥ 10.0 TPM in at least 10% tumors" (Table R9).

Table R9 Genes filtered by initial expression filter

Gene	50% quantile expression/TPM	90% quantile expression /TPM
DNAH9	0.55	3.806
ODF1	0	0.043

On the other hand, all SMGs and genes with significant mutational cluster reported in initial Manuscript had relatively high expression and were not affected by the new expression filter (Table R10).

Table R10 Genes reported in initial Manuscript had relatively high expression

Gene	Type*	90% quantile expression /TPM
TP53	SMG + Mutational cluster	155.134
RB1	SMG	62.794
NOTCH1	SMG + Mutational cluster	27.443
FBXW7	Mutational cluster	62.159
EP300	Mutational cluster	80.015

*SMG: significantly mutated gene

We updated the definition of "expressed in tumors" in the revised Manuscript and reported the 90% quantile of expression in the revised Supplementary Table 5.

Minor comments:

1. The term 'insufficient immune cell infiltration' may better be replaced by 'T cells exclusion', as insufficient have unjustified implications (insufficient for what?), and only T cells were tested. PSCCE may actually be enriched for other immune cell types.

Response:

We appreciate this suggestion. We replaced “insufficient immune cell infiltration” by “T cell exclusion” in the abstract, the tumor microenvironment section and discussion of the revised Manuscript.

2. In mutation identification analysis, it will be informative to add the total number of mutations, not just the nonsynonymous. Synonymous mutations might also have an effect, especially if identified in the same gene in multiple tumors. The reader is left to wonder if the 4713 in L113 is the total number of mutations or just of single nucleotide substitutions and not indels, for example.

Response:

We appreciate this comment. We updated our description by stating (newly added part is underlined):

L105: “We identified 4918 somatic mutations by WES, including 1200 synonymous single nucleotide variants (SNVs) and 3511 nonsynonymous SNVs and 207 indels (Supplementary Table 2)”.

L113: “We observed a high frequency of C>T substitutions, comprising 51.1% (2407/4711) of all SNVs (including 1200 synonymous SNVs and 3511 nonsynonymous SNVs)”.

3. Similarly, in L167-169, the reader is left to wonder how many TP53 were in total, assuming 52 stands for all nonsynonymous mutations.

Response:

We further clarified our description of *TP53* mutations by replacing L167 to L169 with the following statement in revised Manuscript:

“In total, 54 somatic mutations (43 nonsynonymous SNVs, 2 synonymous SNVs and 9 indels) of TP53 were observed, affecting 85% (39/46) of all tumors. Thirty-four percent (18/52) of nonsynonymous mutations of *TP53* were nonsense, frameshifting indels or splice site mutations truncating the protein.”

4. L 107-109 "The mutation rate of PSCCE in this study was significantly higher than that reported by Wang et al12 (P = .0225), owing to the higher sequencing depth." – What was Wang mutation rate? This claim should be supported by subsampling the reads to similar sequencing depth and repeating the analysis.

Response:

In Wang et al study, overall mutation rate was 2.85/Mb and nonsynonymous mutation rate was 2.12/Mb. Our study identified an overall mutation rate of 3.05/Mb and nonsynonymous mutation rate of 2.31/Mb.

Following reviewer’s suggestion, we performed subsampling of 46 sample pairs to 129× using samtools. As expected, overall mutation rate decreased to 2.68/Mb, significantly lower than our original 318× results ($P = 0.0011$, paired Wilcoxon test), but comparable to that reported by Wang et al ($P = 0.4697$, Wilcoxon rank-sum test).

We added the nonsynonymous mutation rate of Wang et al study in our revised Manuscript for a clearer comparison, stating: “The nonsynonymous mutation rate in present study was significantly

higher than that reported by Wang et al (2.12/Mb, $P = 0.0225$, Wilcoxon rank-sum test), owing to higher sequencing depth.”

5. Several methods are not cited appropriately. No reference for MutsigCV, MuTect2, CNVkit, mutational signatures (L536), RSEM, DESEQ, Consensus clustering and more.

Response:

We appreciated this comment. We carefully checked our Manuscript and added proper references to BWA, MuTect2, MutsigCV, SomaticSignatures, CNVkit, DGV, STAR, RSEM, DESeq2, Consensusclusterplus and CIBERSORT.

6. L162-169 described the genes identified as significantly mutated. The L170-176 list other genes that were mutated – how were those identified? If by manual inspection, how many genes were manually inspected? What was the selection criteria? How many of those were mutated?

Response:

By MutsigCV and mutational cluster analysis, we identified *TP53*, *RB1*, *NOTCH1*, *EP300* and *FBXW7* as significantly mutated genes. Detailed mutation of *TP53* was described in L167-L169. *RB1* and Notch pathway suffered dysregulation by multiple mechanism and were described as two independent section.

We manually inspected 97 genes that were mutated in at least three tumors and attempted to identify cancer-associated genes (the COSMIC Cancer Gene Census, ref 1), or genes within same oncogenic pathway (according to ref 2). We found 22 cancer genes with established role in cancer. Histone modifier, including *EP300*, *CREBBP*, *KMT2D* and *KDM6A*, comprised a large fraction of the 22 genes list. They were also reported to be frequently mutated in both SCLC and esophageal cancers. We also found frequent mutation of atypical cadherins, including *FAT1*, *FAT3* and *FAT4* and presented them as part of Fig 1e.

For the conciseness of the manuscript, we did not describe number of mutation in details for each gene. Instead, full mutation list were provided for readers to search for genes of interest.

We updated our description in the revised Manuscript by stating: “We sought genes with established roles in cancers (the Cancer Gene Census, ref.24) or genes within same oncogenic pathway in 97 genes that were mutated in at least 3 tumors. We observed frequent mutations in histone modifiers...”

Ref 1, Sondka et al. Nat Rev Cancer. 2018;18(11):696-705.

Ref 2, Sanchez-Vega et al. Cell. 2018;173(2)321-337.

7. L182 – homozygous deletion of a gene is by definition mutually exclusive with a mutation in that gene. Possibly partial deletion?

Response:

Yes, 13 out of 14 *RB1* homozygous deletions observed in PSCCE are partial deletion, affecting a median of 11 exons (range: 1 to 25) but not the whole *RB1* locus and having a median length of 52.5kb

(range: 0.2kb to 2828kb). We used “exon deletions” to replace the expression “homozygous deletion of *RB1*” in the revised Manuscript. Table R11 below listed all the exon deletions observed in RB1 (more than initially reported because we revised our CNV calling pipeline following comment from Reviewer #1).

Table R11 RB1 exon deletion events observed in PSCCE

#	Sample	Genomic Ranges	Length/kb	Affected exons
1	PSCCE_10T	chr13: 48916724-48955589	38.9	Exon 3 to exon 17
2	PSCCE_12T	chr13: 49030345-49030538	0.2	Exon 19
3	PSCCE_13T	chr13: 48916724-48955589	38.9	Exon 3 to exon 17
4	PSCCE_14T	chr13: 49047412-49047640	0.2	Exon 24
5	PSCCE_15T	chr13: 48985562-49109906	124.3	Exon 18 to exon 27 of RB1 Plus part of downstream RCBTB2 gene
6	PSCCE_16T	chr13: 48934140-48986599	52.5	Exon 7 to exon 17
7	PSCCE_29T	chr13: 49027124-49054249	27.1	Exon 18 to exon 27
8	PSCCE_31T	chr13: 48956089-49089882	133.8	Exon 18 to exon 27 of RB1 Plus part of downstream RCBTB2 gene
9	PSCCE_40T	chr13: 48934140-49051603	117.5	Exon 7 to exon 26
10	PSCCE_47T	chr13: 49027124-51855425	2828.3	Exon 3 to exon 27 of RB1 Plus downstream numerous genes
11	PSCCE_61T	chr13: 48916724-48985062	68.3	Exon 3 to exon 17
12	PSCCE_65T	chr13: 48934140-49106586	172.4	Exon 7 to exon 27 of RB1 Plus part of downstream RCBTB2 gene
13	PSCCE_77T	chr13: 49037870-49039526	1.7	Exon 21 to exon 23
14	PSCCE_79T	chr13: 48807471-50881863	2074.4	Full length of RB1 Plus downstream numerous genes

Hence, in tumors which carried *RB1* exon deletions, part of *RB1* gene body pertained in genome. The remaining exons were theoretically susceptible to SNVs or indels. However, we discovered no SNVs or indels in *RB1* gene of these tumors.

8. L223,262 – differential expression analysis of PSCCE against what? Matched normal samples? Of which tissue? Other cancer types?

Response:

In differentially expressed gene (DEG) analysis, we compared the 38 tumors against 23 matched normal esophageal samples sequenced together in our study.

We clarified this process in the revised Manuscript, by stating (newly added part is underlined):

“We next turned to transcriptomic landscape of PSCCE...We compared gene expressions of 38 PSCCE tumors against 23 match normal esophageal samples to identify significantly differentially expressed genes (DEGs).”

The corresponding Method section has also been updated to: “Differentially expressed genes (DEGs) were identified using R package “DESeq2”, by comparing 38 tumors against 23 matched normal

esophageal samples sequenced together in the present study.”

9. L232 seed should be see

Response:

Thank you for pointing this out. We corrected the word in revised Manuscript.

10. Figure 3c – labels seem to be misplaced.

Response:

Yes, these labels belonged to Fig 3f and were misplaced there. We removed these labels in the revised Manuscript.

11. Figure 3g.g – what is the order of the genes?

Response:

Signature “RB_loss signature” was obtained from Adam Ertel et al, 2010. *Cell Cycle*, 9:20, 4153-4163. This signature contained 159 genes which were upregulated upon *RB1* deletion.

The whole list of genes are as following:

AEBP2, ANAPC11, ANAPC5, ANGPTL2, ANLN, ANP32B, ARHGAP21, ASF1B, ATM, BIRC5, BRCA1, BRCA2, BRRN1, BUB1, CASP8AP2, CBX2, CBX5, CCNA2, CCNB1, CCNB2, CCNF, CD34, CDC20, CDC25C, CDC45L, CDC6, CDCA3, CDCA5, CDCA7, CDCA8, CDK2, CDKN1C, CDKN3, CENPA, CHAF1B, CHEK1, CKAP2, CKS2, DCK, DDIT4, DEK, DLG7, DNAJC9, DNMT1, DOK1, DTYMK, E2F1, ECT2, EGR1, EI24, EIF4A2, ERCC5, ETV4, EZH2, FBLN1, FEN1, FEZ1, FOXM1, GMNN, GTSE1, H2AFZ, HAT1, HELB, HIP1R, HMGA2, HMGB2, HMGB3, HMGN1, HMGN2, HNRPC, HNRPD, HNRPR, HNRPU, INCENP, KIF11, KIF1C, KIF20A, KIF22, KIF23, KIF2C, KLF4, KPNA2, LBR, LIG1, MAD2L1, MCM2, MCM3, MCM4, MCM5, MCM7, MDM2, MKI67, MRE11A, MSH2, MTCP1, NAP1L1, NASP, NEK2, NUSAP1, ORC6L, PCNA, PDCD6IP, PDGFRA, PERP, PHC1, PHF13, PLK1, PLK4, PLTP, PML, POLD1, PRC1, PRDX4, PRIM1, PSIP1, RACGAP1, RAD21, RAD51, RAD51AP1, RBBP4, RBL1, REV3L, RFC2, RFC5, RGENEF, RPA1, RRM1, RRM2, SIVA, SLBP, SLK, SMC2L1, SMC4L1, STMN1, TACC3, TARDBP, TCF19, TFDP1, TK1, TMPO, TOP2A, TOPBP1, TRIP13, TTK, TUBGCP3, TYMS, UHRF1, UPF3B, USP1, WAC, WDHD1, WIG1, XRCC1, AURKA, AURKB, CCNE1, CCNE2, CDT1, DBF4

Full citation of Ertel A et al. paper was provided in the Manuscript.

12. L278 and elsewhere – consider replacing changeable by unstable.

Response:

We replaced “changeable” by “unstable” in revised Manuscript.

13. L289 'integrated analysis revealed' – what type of analysis? How significant is the association?

Response:

After two subtypes were identified, we attempt to find SCNVs associated with subtypes. SCNVs that affected ≥ 3 of 38 RNA-sequenced tumors were collected. Distribution of SCNVs were compared between two subtypes, using Fisher’s exact test with a significant level of 0.05. We discovered 255 gene amplifications (including *MYC* amplification) associated with PSCCE-N subtype, no gene amplification associated with PSCCE-A subtype and no deletions showing significant difference. We did not perform multiple comparison correction because individual test was not independent: genes

proximal to each other on the genome tended to change together as a segment. Instead, we perform chromosomal location enrichment of significant SCNVs. SCNVs enriched in specific genome loci suggest biological relevance but not significant p values due to *pure chance*.

The 255 gene amplifications associated with PSCCE-N were supplied as input for chromosomal location enrichment on Enrichr (<http://amp.pharm.mssm.edu/Enrichr/>). We observed significant enrichment for chromosomal location of chromosome 8q segments, including *MYC* on chr8q24.

We updated our Method in revised Manuscript, by stating: “SCNVs that affected ≥ 3 of 38 RNA-sequenced tumors were tested for significantly different distribution between two subtypes using Fisher’s exact test. Chromosomal location enrichment analysis of SCNVs with $p < 0.05$ was performed on Enrichr (<http://amp.pharm.mssm.edu/Enrichr/>). Chromosomal locations with $p < 0.01$ and $q < 0.05$ were considered significant.”

14. L293, 294 – what test and significance level were used?

Response:

MYC, *MYCL*, *MYCN* mRNA level were compared between two subtypes using Wilcoxon rank-sum test. We clarified this in revised Manuscript by stating the following: “However, mRNA level of *MYC* and *MYCL* showed no significant difference between two subtypes. PSCCE-N tumors had a significantly higher *MYCN* level than PSCCE-A tumors ($P = 0.0012$, Wilcoxon rank-sum test, Supplementary Fig 4b).”

15. L308 – consistent with which of the findings described before?

Response:

We clarified our conclusion by stating (newly added part is underlined): “The SCLC-N subtype was reported to associate with *MYC* amplification and to rapidly metastasize and relapse in human and murine model, consistent with our findings that PSCCE-N subtype was associated with *MYC* amplification and poorer prognoses.”

16. L322, 324 – significant by which test? P-value?

Response:

NOTCH2, *NOTCH3*, *HES1* and *ASCL1* were identified as significantly differentially expressed genes (DEGs) by DESeq2 package. DESeq2 used Wald test to calculate *P* value and Benjamini-Hochberg procedure to estimate *q* value. We reported genes with $\log_2(\text{Foldchange}) > 1$, $P < 0.01$, $q < 0.05$ as significantly upregulated genes in and genes with $\log_2(\text{Foldchange}) < -1$, $P < 0.01$, $q < 0.05$ as significantly downregulated. We added the thresholds to the revised Method section.

We provided a new Source Data 3 file of all DEGs with their $\log_2(\text{Foldchange})$, *P* value and *q* value to the readers. For the conciseness of text, we did not list *P* values, *q* values and $\log_2\text{FC}$ for each gene. We invite the readers to this supplementary file, by stating (newly added part is underlined): “Relatively mutation-sparse *NOTCH2* and *NOTCH3* genes were identified as downregulated DEGs when compared to matched normal samples (Fig 5d, detailed *P*, *q* and $\log_2\text{FC}$ values in Source Data 3).”

17. L334-335 – where is this finding shown?

Response:

Upregulation of *DLL3* was presented as part of Fig 5d of initial Manuscript. We added boxplot of *DLK1* mRNA expression to Fig 5d in revised Manuscript.

We also provided a Source Data 3 of all DEGs. We updated text, stating (newly added part is underlined): “Transcriptomic profiling also revealed inhibitory ligands of Notch pathway *DLK1* and *DLL3* as upregulated DEGs (Fig 5d, also see Source Data 3).”

18. Figure 5d – what do N and T stand for?

Response:

We appreciate this comment. T stands for “Tumor” and N stands for “Normal”. This information was added to legend of Fig 5d in the revised Manuscript.

19. L357 Projection on what? The PSCCE RNA-seq data?

Response:

We performed projection on quantile normalized expression matrix of PSCCE (n = 38), SCLC (n = 81), LUAD (n = 81), LUSC (n = 81), EAC (n = 38) and ESCC (n = 38), which was same to the gene expression matrix we used in analysis related to Fig 3.

We clarified this in the revised Manuscript, by stating (newly added part is underlined):” We first applied single-sample GSEA (ssGSEA) projection of immune cell signatures collected from the literatures (n = 19, Supplementary Table 14), including signatures of T cells, B cells, macrophages and granulocytes, on gene expression profiles of PSCCE, SCLC, LUAD, LUSC, EAC and ESCC (see Method and Source Data 4).”

We updated related Method section, by stating: “Gene expression profiles of PSCCE, SCLC, LUAD, LUSC, EAC and ESCC samples in Supplementary Table 9 were combined and quantile normalized.”

20. L362 Deconvolution does not confirm signature projection, merely repeat the analysis – those are two computational ways to show the same thing based on the same data.

Response:

Thank you for pointing this out. The description in initial Manuscript was ambiguous. L362 was replaced with the following sentence “Enumeration of immune infiltrates using a deconvolution method (CIBERSORT, ref.36) showed similar trend” in the revised Manuscript.

21. L388 filed should have probably be field

Response:

Thank you for pointing this out. We corrected this word in revised Manuscript.

22. L479 ethnic should probably be ethic

Response:

Thank you for pointing this out. We corrected this word in revised Manuscript.

23. L536-L539 – Consider replacing the last two sentences of the paragraph.

Response:

We appreciate this comment. We revised the last two sentences to better describe the mutational signature analysis process, by stating (revised part is underlined): “For each K , the cosine similarities between resultant signatures and COSMIC signatures (version 2) were calculated. Optimal K was determined when (i) explained variance did not increase remarkably by further increasing K , and (ii) the average of cosine similarities to the most resembling COSMIC signatures reached maximum.”

24. L545 – RNA sequencing description is not detailed enough – read length and paired or non paired reads should be specified.

Response:

We appreciate this comment. We further clarified this in the Method section of revised Manuscript, by stating (newly added part is underlined): “Paired-end 150bp sequencing of the subsequent libraries were performed on Illumina NovaSeq platform by WuXi NextCODE (Shanghai), with the aim to obtain ≥ 6 gigabytes (Gb) sequencing data.”

25. L548 – what makes a sequence read validated? How many of the reads were not validated?

Response:

RNA sequencing reads were checked by Trimmomatic (0.36, Ref 1) using parameter “ILLUMINACLIP:TruSeq3-PE-2.fa:2:30:10:8:true MINLEN:75”. Briefly, a validated reads had mismatch count < 3 bp, read length > 75 bp and accuracy of matching above a given threshold.

By Trimmomatic trimming, a median of 99.57% (range: 97.58% to 99.78%) of raw reads were kept as validated reads. Thus, only very small proportion of reads were not validated.

We updated RNA-seq data processing in the Method section of revised Manuscript, stating (newly added part is underlined): “Sequencing reads were trimmed by Trimmomatic (0.36) and aligned to reference genome hg19 by STAR (2.4.2a).”

Ref 1: Bolger et al. *Bioinformatics*. 2016;30(15):2114-2120

26. In several places (L594, L516, L609 and elsewhere), p-value threshold is stated as 0.05, and though multiple comparisons were done, no correction is mentioned. State type of correction if performed, or perform correction if not performed.

Response:

We thank the Reviewer for pointing this out.

In L594 (identification of signature genes of three groups) of original Manuscript, we calculated P value using Tukey-HSD test. Following the reviewer’s suggestion, we estimated false discovery rate (FDR) through Benjamini-Hochberg (BH) procedure in the revised Manuscript. We then required a signature gene to have P value < 0.05 and $\log_2FC \geq 2$ and $FDR < 0.1$. We found no signature genes in the original Manuscript was rejected. In fact, the maximal FDR for signature genes of each group were well below the threshold (Table R12).

Table R12 Maximal estimated FDR in signature genes of each group

Group	Maximal FDR of signature genes
Group NE	3.69×10^{-8}
Group Adeno	1.113×10^{-4}
Group Squamous	1.975×10^{-8}

Similarly, we estimated FDR corresponding to P value of each tests described in L609 (identification of genes of PSCCE subtypes) through BH procedure. We required a signature gene to have P value < 0.05 and $\log_2FC \geq 2$ and q value < 0.1 . We found in signature genes of PSCCE-A subtype, 6 out of 115 were rejected; in signature genes of PSCCE-N, 3 out of 36 were rejected. The rejected genes were listed in Table R13. None of them was involved in downstream analysis.

Table R13 Subtype signature genes that were excluded due to non-significantly q value

Subtype	Gene	Log ₂ (FC)	P value	q value
PSCCE-A	SLITRK6	2.3503	0.00686	0.10637
	KLK12	2.2377	0.01443	0.16231
	IGLC7	2.2593	0.01650	0.17417
	IGHV1-2	2.2366	0.01194	0.14674
	IGKV1-12	2.0422	0.01915	0.18855
	TCEB3CL2	2.2407	0.0120	0.14729
PSCCE-N	PCDH11X	2.3126	0.0285	0.23216
	NELL2	2.2920	0.01161	0.14423
	DCC	2.0253	0.0069	0.10759

For L516, we estimated FDR level from pval_tn values (Supplementary Table S2, related to filter comparing variant reads distribution in tumor and normal) of all raw Mutect2 calls before any of the seven filters was applied. We found that, even by very conservative BH procedure estimation, a p value of 0.05 corresponded to an FDR of only 0.0931 (< 0.1). By applying a series of seven filters, FDR in final list of somatic mutations would be well below 0.1.

We clarified the correction processes applied to L594, L609 and L516, and updated the Method section. Benjamini-Hochberg FDR q values were added to Supplementary tables 2, 10 and 12.

27. L608 what filtering was done prior to differential expression analysis?

Response:

The input for discovery of subtype signatures were genes with an average expression ≥ 1.0 TPM. In total, expressions of 18238 genes were compared between two subtypes.

We clarified this in the revised Manuscript by stating: “To better discover signature genes for each subtype, we compared all genes with an average expression ≥ 1.0 TPM”.

REVIEWERS' COMMENTS:

Reviewer #1 (Remarks to the Author):

The authors have done an adequate job addressing comments and have improved the clarity of the report.

Reviewer #2 (Remarks to the Author):

The investigators have done a good job addressing my initial concerns and suggestions regarding this manuscript.

Charles Rudin

Reviewer #3 (Remarks to the Author):

The authors reasonably addressed all comments.

We appreciate the time and efforts of the Reviewers. Please find our response below.

Reviewer #1 (Remarks to the Author):

The authors have done an adequate job addressing comments and have improved the clarity of the report.

Response:

We thank the comments from Reviewer #1 which helped us improve the SCNV calling pipeline.

Reviewer #2 (Remarks to the Author):

The investigators have done a good job addressing my initial concerns and suggestions regarding this manuscript.

Charles Rudin

Response:

We thank Dr. Rudin for his helpful comments on detailed description of PSCCE subtypes..

Reviewer #3 (Remarks to the Author):

The authors reasonably addressed all comments.

Response:

We thank Reviewer #3 for the helpful comments on RNA-seq analysis.